# RETHINKING THE ROLE OF FRAMES FOR SE(3)-INVARIANT CRYSTAL STRUCTURE MODELING

**Yusei Ito**[*1,2]**, Tatsunori Taniai**[*1]**, Ryo Igarashi**[1]**, Yoshitaka Ushiku**[1]**, and Kanta Ono**[2]
[*]Contributed equally. [1]OMRON SINIC X Corporation [2]Osaka University
`https://omron-sinicx.github.io/crystalframer/`

## ABSTRACT

Crystal structure modeling with graph neural networks is essential for various applications in materials informatics, and capturing SE(3)-invariant geometric features is a fundamental requirement for these networks. A straightforward approach is to model with orientation-standardized structures through structure-aligned coordinate systems, or "frames." However, unlike molecules, determining frames for crystal structures is challenging due to their infinite and highly symmetric nature. In particular, existing methods rely on a statically fixed frame for each structure, determined solely by its structural information, regardless of the task under consideration. Here, we rethink the role of frames, *questioning whether such simplistic alignment with the structure is sufficient*, and propose the concept of *dynamic frames*. While accommodating the infinite and symmetric nature of crystals, these frames provide each atom with a dynamic view of its local environment, focusing on actively interacting atoms. We demonstrate this concept by utilizing the attention mechanism in a recent transformer-based crystal encoder, resulting in a new architecture called CrystalFramer. Extensive experiments show that CrystalFramer outperforms conventional frames and existing crystal encoders in various crystal property prediction tasks.

## 1 INTRODUCTION

Geometric graph neural networks (Xie & Grossman, 2018; Chen et al., 2019; Choudhary & DeCost, 2021; Lin et al., 2023), including transformer variants (Yan et al., 2022; 2024; Taniai et al., 2024), play a central role in machine learning (ML)-based structural modeling of materials. This technology offers a powerful alternative to conventional simulation methods, such as density functional theory (DFT) calculations, enabling high-throughput prediction of material properties. Furthermore, it serves as the basis for various ML applications in materials science, such as material embedding learning (Suzuki et al., 2022; 2025; Chiba et al., 2023) and crystal generation (Jiao et al., 2023).

A key requirement for these networks is the ability to capture essential features of materials embedded in their crystal structures. Crystal structures are periodic, infinitely repeating arrangements of atoms in 3D space, typically represented by minimum repeatable patterns called unit cells. Material properties, such as formation energy and bandgap, are invariant under rigid transformations (*i.e.*, rotations and translations) in crystal structures, as well as under variations in their unit cells. This fact leads to the so-called periodic SE(3) invariance (Yan et al., 2022) as an essential property for crystal encoders. Therefore, recent studies have explored various forms of richer yet invariant structural information beyond the simplest interatomic distances (Chen & Ong, 2022; Duval et al., 2023; Yan et al., 2024).

One approach that has shown promising results for molecules (Puny et al., 2022) is the use of "frames." A frame is a coordinate system aligned equivariantly to a given structure to provide an orientation-standardized view of the structure (see Fig. 1, left). Frames allow arbitrary networks to directly exploit rich 3D structural features, including the relative positions between atoms and their directions, without imposing any architectural constraints. However, determining frames for crystals is more challenging than for molecules, primarily due to the infinite and symmetric nature of crystals.

In this work, we study a new family of frames for crystal structures in rethinking the role of frames. We hypothesize that *the essential role of frames is not merely to provide a structure-aligned coordinate system for a given structure, but rather to align the coordinate system with the interatomic interactions*

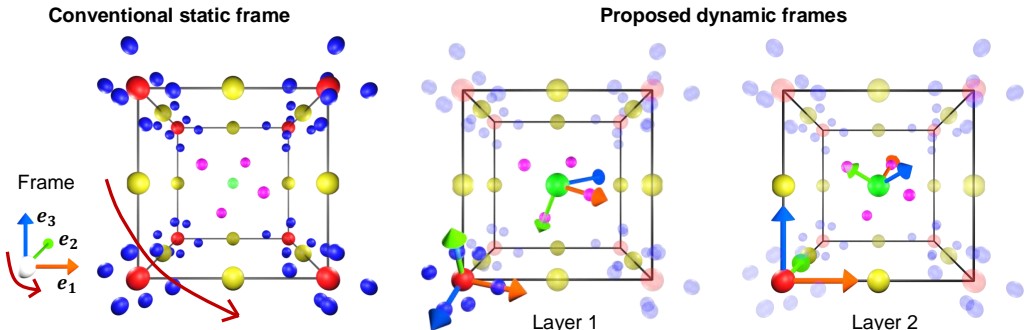

**Figure 1: Conventional static frame and proposed dynamic frames.** Conventional frames are determined statically to align with the structure, ensuring consistency under rotation and providing a canonical global representation of the structure. This consistency is schematically illustrated by the curved arrows. By contrast, the proposed dynamic frames are determined for each atom in each message-passing layer, by considering the local dynamic environment around that atom in that layer.

*acting on the structure.* Following this belief, we propose a novel concept of *dynamic frames*. These frames define local coordinate systems centered on individual atoms by dynamically accounting for the atoms actively engaged in learned interactions in each interatomic message-passing layer (Fig. 1, right). This concept challenges the conventional notion of 'static frames,' which are based on the premise of providing fixed views of structures (Puny et al., 2022). Thus, whether such a dynamic frame is effective or not is an unexplored non-trivial question, which we aim to answer.

To verify this concept, we develop several types of dynamic frames by utilizing the self-attention mechanism (Taniai et al., 2024) to quantify the interaction engagement. We conduct extensive comparisons on datasets derived from the JARVIS, Materials Project (MP), and Open Quantum Materials Database (OQMD). Our results show that our method outperforms existing frame methods for crystals (Duval et al., 2023; Yan et al., 2024) and other state-of-the-art networks (Choudhary & DeCost, 2021; Chen & Ong, 2022; Yan et al., 2022; 2024; Lin et al., 2023; Taniai et al., 2024) across various crystal property prediction tasks. We release our code online.

## 2 PRELIMINARIES

### 2.1 CRYSTAL STRUCTURE

A crystal structure is described by its 3D unit cell slice, denoted as $(A, P, L)$ following Yan et al. (2022). A unit cell is a parallelepipedal structure containing a finite number, say $N$, of atoms. The species (atomic numbers) and 3D Cartesian coordinates of these atoms are provided as $A = [a_1, a_2, ..., a_N] \in \mathbb{N}^{1 \times N}$ and $P = [\boldsymbol{p}_1, \boldsymbol{p}_2, ..., \boldsymbol{p}_N] \in \mathbb{R}^{3 \times N}$. The parallelepipedal cell shape is given by three vectors: $L = [\boldsymbol{l}_1, \boldsymbol{l}_2, \boldsymbol{l}_3] \in \mathbb{R}^{3 \times 3}$, called lattice vectors. By tiling the parallelepiped unit cell to fill 3D space, the species and positions of all the atoms in the crystal structure are determined as

$$\hat{A} = \{a_{i(\boldsymbol{n})} | a_{i(\boldsymbol{n})} = a_i, \boldsymbol{n} \in \mathbb{Z}^3, 1 \leq i \leq N\}, \tag{1}$$

$$\hat{P} = \{\boldsymbol{p}_{i(\boldsymbol{n})} | \boldsymbol{p}_{i(\boldsymbol{n})} = \boldsymbol{p}_i + L\boldsymbol{n}, \boldsymbol{n} \in \mathbb{Z}^3, 1 \leq i \leq N\}. \tag{2}$$

Following Taniai et al. (2024), we use $i$ to denote the $i$-th atom in a unit cell, and $i(\boldsymbol{n})$ to denote its duplicate by a unit-cell translation: $L\boldsymbol{n} = n_1\boldsymbol{\ell}_1 + n_2\boldsymbol{\ell}_2 + n_3\boldsymbol{\ell}_3$. We use $j$ and $j(\boldsymbol{n})$ similarly.

### 2.2 TRANSFORMERS FOR CRYSTAL STRUCTURES

Geometric graph neural networks are used as crystal encoders in various materials-related tasks. These encoders typically represent the state of a given crystal structure by a set of atom-wise abstract state features, $X = [\boldsymbol{x}_1, \boldsymbol{x}_2, ..., \boldsymbol{x}_N] \in \mathbb{R}^{d \times N}$. These states are initially provided as atom embeddings, $X^{(0)} \leftarrow \text{AtomEmbedding}(A)$, which only symbolically represent atomic species. The encoders then evolve these states through interatomic message-passing layers, $X^{(t+1)} \leftarrow f^t(X^{(t)}, P, L)$, to

eventually reflect the atomic states in the given structure appropriate for a target task. Since the seminal work of Xie & Grossman (2018) and Schütt et al. (2018), graph neural networks (GNNs) have long been the standard for crystal encoders until the advent of transformer-based networks by recent work (Yan et al., 2022; 2024; Taniai et al., 2024).

In particular, Taniai et al. (2024) have developed simple physics-informed formalism for crystal encoders using a self-attention mechanism. By imitating interatomic potential summations for energy calculations in physics simulations, they model the evolution of current state $\boldsymbol{x}$ using *infinitely connected distance-decay attention*. This attention mechanism models the interactions between each unit-cell atom $i$ and all the infinitely repeating atoms $j(\boldsymbol{n})$ in the entire crystal structure as

$$\boldsymbol{x}_i' = \frac{1}{Z_i} \sum_{j=1}^{N} \sum_{\boldsymbol{n} \in \mathbb{Z}^3} \exp \left( \frac{\boldsymbol{q}_i^T \boldsymbol{k}_j}{\sqrt{d_K}} - \frac{\|\boldsymbol{p}_{j(\boldsymbol{n})} - \boldsymbol{p}_i\|^2}{2\sigma_i^2} \right) \left( \boldsymbol{v}_j + \boldsymbol{\psi}_{ij(\boldsymbol{n})} \right). \tag{3}$$

Here, query $\boldsymbol{q}$, key $\boldsymbol{k}$, and value $\boldsymbol{v}$ are linear projections of current state $\boldsymbol{x}$. Scalar $\sigma_i$ is a tail-length variable for Gaussian distance-decay attention, adaptively derived from $\boldsymbol{x}_i$. Vector $\boldsymbol{\psi}_{ij(\boldsymbol{n})}$ is a geometric position embedding that encodes the distance, $\|\boldsymbol{p}_{j(\boldsymbol{n})} - \boldsymbol{p}_i\|$, between atoms $i$ and $j(\boldsymbol{n})$. Scalar $Z_i = \sum_j \sum_{\boldsymbol{n}} \exp(\boldsymbol{q}_i^T \boldsymbol{k}_j / \sqrt{d_K} - \|\boldsymbol{p}_{j(\boldsymbol{n})} - \boldsymbol{p}_i\|^2 / 2\sigma_i^2)$ is the normalizer of softmax attention weights. The exponential distance-decay factor in Eq. 3 provably ensures its rapid convergence within a finite range of unit-cell shifts $\boldsymbol{n}$ (Taniai et al., 2024).

Their method, called Crystalformer, enjoys a good balance between a strong physically-motivated inductive bias and the flexibility of abstract feature representations. It is considered the state of the art with other GNN-based (Lin et al., 2023) and transformer-based (Yan et al., 2024) methods.

We utilize Crystalformer as a baseline in this work. This is because its architecture closely follows the standard softmax attention (Vaswani et al., 2017) and is suitable to demonstrate our concept of dynamic frames, while other existing transformers (Yan et al., 2022; 2024) use distinct channel-wise sigmoid attention. We discuss this more in Sec. 6. Our method, described in Sec. 3, extends position embedding $\boldsymbol{\psi}_{ij(\boldsymbol{n})}$ in Eq. 3 to incorporate richer, invariant information beyond distances $\|\boldsymbol{p}_{j(\boldsymbol{n})} - \boldsymbol{p}_i\|$.

### 2.3 FRAMES FOR SE(3)-INVARIANT STRUCTURAL MODELING

**Frame averaging.** Puny et al. (2022) have introduced Frame Averaging (FA) as a general framework to adapt networks to be invariant (or equivariant) to certain symmetries in input data. Although FA is originally grounded in group representation theory, we provide a high-level review focused on SE(3)-invariant modeling of 3D point clouds. Given a point cloud as $P$, FA computes a frame, $F \in \mathcal{F}(P)$, as a coordinate system inherent to and aligned with $P$ (Fig. 1, left). For instance, $\mathcal{F}$ is principal component analysis (PCA) applied to $P$. Each frame $F$ thus defines a geometric transformation that maps $P$ to a canonical, rotation-invariant representation, $FP$. However, $\mathcal{F}(P)$ may not uniquely provide a single frame due to algorithmic ambiguities in $\mathcal{F}$ or symmetries in $P$. Even in such cases, FA allows us to derive rotation-invariant (*i.e.*, SO(3)-invariant) networks $\bar{f}_{\mathcal{F}}$ from any given networks $f$, by averaging $f$'s outputs over all possible finite frames, as follows:

$$\bar{f}_{\mathcal{F}}(X, P) = \frac{1}{|\mathcal{F}(P)|} \sum_{F \in \mathcal{F}(P)} f(X, FP). \tag{4}$$

The translation invariance is further attained by formulating $f$ with relative positions (*e.g.*, $F\boldsymbol{p}_j - F\boldsymbol{p}_i$), bringing SE(3) invariance to $\bar{f}_{\mathcal{F}}$. FA can powerfully adapt arbitrary networks to be SE(3) invariant without constraining the architectural design. However, it hinders efficiency, as the computation increases with the number of possible frames. Stochastic FA by Duval et al. (2023) mitigates this issue by randomly selecting a single frame from $\mathcal{F}(P)$ during training. This scheme enforces networks $f$ to learn the invariance to frame variations, approximately achieving SE(3) invariance.

**PCA frames.** Puny et al. (2022) originally applied FA for molecules using PCA-based frames, and Duval et al. (2023) later extended it for crystals by simply treating unit cell structures $P$ as finite-sized point clouds. These PCA frames compute three orthogonal eigenvectors $\{\boldsymbol{e}_1, \boldsymbol{e}_2, \boldsymbol{e}_3\}$ of the covariance matrix for $P$, corresponding to eigenvalues $\lambda_1 \geq \lambda_2 \geq \lambda_3$, as the frame axes: $F = [\boldsymbol{e}_1, \boldsymbol{e}_2, \boldsymbol{e}_3]^T$. Because of the sign ambiguity of the eigenvectors, PCA produces eight frames for O(3)/E(3) invariance and four frames for SO(3)/SE(3) invariance with the restriction of $\det(F) = 1$.

Although PCA is well-established, it suffers from eigenvalue degeneration for highly symmetric data, such as crystal structures. For example, PCA for cubes produces identity covariance matrices up to a constant scale, whose eigenvectors are arbitrary vectors $\boldsymbol{e} \in \mathbb{R}^3$. The crystal frame construction by Duval et al. (2023) is thus vulnerable to this degeneration issue and, moreover, sensitive to unit-cell variations of the same crystal structure.

**Lattice frames.** Yan et al. (2024) have proposed frames based on the lattice vectors of crystals, as similar to *reduced cells* (*i.e.*, uniquely determined minimum cells) (Niggli, 1928). Specifically, their method selects a lattice point, $\boldsymbol{e} = n_1 \boldsymbol{\ell}_1 + n_2 \boldsymbol{\ell}_2 + n_3 \boldsymbol{\ell}_3$, with the minimum non-zero norm $\|\boldsymbol{e}\|_2$ as first axis $\boldsymbol{e}_1$, and selects the second and third smallest ones as axes $\boldsymbol{e}_2$ and $\boldsymbol{e}_3$ while ensuring $\text{rank}(\boldsymbol{e}_1, \boldsymbol{e}_2, \boldsymbol{e}_3)$ is full. The signs of these axes are adjusted so that the angles between $\boldsymbol{e}_1$ and $\boldsymbol{e}_2$ and between $\boldsymbol{e}_1$ and $\boldsymbol{e}_3$ become acute and the coordinate system is right-handed (*i.e.*, $\det(F) > 0$).

Notice that these existing frame methods for crystals, specifically PCA and lattice frames, all provide a statically fixed frame for each crystal structure. Also, both rely on unit cell representations (either points $P$ or lattice vectors $L$), which are rather artificially-introduced crystal descriptions that may not necessarily reflect the physical properties of materials (see Appendix A for more discussion). These observations motivate us to propose the concept of dynamic frames, as we discuss next.

## 3 DYNAMIC FRAMES

In the search for effective frames for crystals, we challenge the conventional notion of frames, which implicitly follows the simple premise of representing structures in a canonical manner (Puny et al., 2022; Duval et al., 2023; Yan et al., 2024). Let us reconsider how frames work in GNNs, whose interatomic message-passing layers are assumed to include the following general operation:

$$\boldsymbol{x}_i' = \sum_{j=1}^{N} \sum_{\boldsymbol{n} \in \mathbb{Z}^3} w_{ij(\boldsymbol{n})} \boldsymbol{f}_{i \leftarrow j(\boldsymbol{n})}(\boldsymbol{x}_{j(\boldsymbol{n})}, \hat{P}). \tag{5}$$

This equation describes that state $\boldsymbol{x}_i$ of each unit-cell atom $i$ is evolved through abstract influences or *messages*, $\boldsymbol{f}_{i \leftarrow j(\boldsymbol{n})}$, from atoms $j(\boldsymbol{n})$ in the crystal structure, with scaling weights $w_{ij(\boldsymbol{n})}$. In standard GNNs (Xie & Grossman, 2018), these weights are pre-defined as neighborhood graphs with a cut-off radius. In recent transformer architectures, the weights are determined dynamically via self-attention, with (Yan et al., 2022; 2024) or without (Taniai et al., 2024) relying on an explicit cut-off radius.

The role of frames in Eq. 5 is to offer, for the design of $\boldsymbol{f}_{i \leftarrow j(\boldsymbol{n})}$, more informative invariant edge features than distances through frame-projected coordinates $F\hat{P}$. From this perspective, constructing a shared frame for the state updates of all atoms $i$, as done in conventional methods, is not preferable. This is because the frame construction can be influenced even by atoms $j(\boldsymbol{n})$ with zero weights in Eq. 5. In other words, when updating the state of atom $i$ in Eq. 5, this atom has its own partial and local view of the entire crystal structure, $\hat{P}$, where weights $w_{ij(\boldsymbol{n})}$ act as a mask on the structure.

This interpretation leads to the concept of *dynamic frames*. That is, we define frames locally for each atom $i$ to align with its interatomic interactions dynamically acting on the structure, rather than directly aligning with the structure itself. We denote these dynamic atom-wise frames as $F_i$. Each $F_i$ is determined based on the masked view of structure $\hat{P}$ with weights $w_{ij(\boldsymbol{n})}$, by emphasizing or de-emphasizing the presence of atoms $j(\boldsymbol{n})$ with larger or smaller weights. Thus, these frames $F_i$ change dynamically depending on target atoms $i$ and also on the layers in a GNN, as shown in Fig. 1.

We hypothesize that dynamically adapting frames for each atom $i$ in each message-passing layer (Eq. 5) provides better invariant edge features via projected coordinates $F_i \hat{P}$. We also point out that these frames are defined with the entire crystal structure, $\hat{P}$, reconstructed from $(P, L)$. This fact highlights an advantage of our frames being invariant to unit cell variations within the same structure.

### 3.1 FRAME DEFINITIONS

We now present several instances of this new family of frames. These frames $F_i$ are constructed for each target atom $i$ in each message-passing layer (Eq. 5), by using coordinates $\hat{P}$ and weights $w_{ij(\boldsymbol{n})}$ of atoms $j(\boldsymbol{n})$ in the structure. We typically assume $w_{ij(\boldsymbol{n})} \geq 0$, but we can use real-valued

weights, for example, by using their absolute values for frame construction. For brevity, we denote $r_{ij(\boldsymbol{n})} = \|\boldsymbol{p}_{j(\boldsymbol{n})} - \boldsymbol{p}_i\|_2$ and $\bar{\boldsymbol{r}}_{ij(\boldsymbol{n})} = (\boldsymbol{p}_{j(\boldsymbol{n})} - \boldsymbol{p}_i)/r_{ij(\boldsymbol{n})}$, both derived from $\hat{P}$.

**Weighted PCA frames.** The first instance of dynamic frames extends the original PCA frames (Puny et al., 2022; Duval et al., 2023). For each target atom $i$ in each message-passing layer, we compute a $3 \times 3$ weighted covariance matrix, $\Sigma_i = \sum_j \sum_{\boldsymbol{n}} w_{ij(\boldsymbol{n})} \bar{\boldsymbol{r}}_{ij(\boldsymbol{n})} \bar{\boldsymbol{r}}_{ij(\boldsymbol{n})}^T$, and computes its orthogonal eigenvectors $\{\boldsymbol{e}_1, \boldsymbol{e}_2, \boldsymbol{e}_3\}$ as the frame axes: $F_i = [\boldsymbol{e}_1, \boldsymbol{e}_2, \boldsymbol{e}_3]^T$. For the sign ambiguity of eigenvectors, we adopt the stochastic FA (Duval et al., 2023) and generate a single frame by randomly flipping the signs of these vectors while ensuring $\det(F_i) = 1$. However, there remains another possible ambiguity in this weighted PCA scheme owing to eigenvalue degeneration by symmetries[1].

**Max frames.** To avoid the degeneration of PCA, we also propose directly selecting atoms $j(\boldsymbol{n})$ with large weights $w_{ij(\boldsymbol{n})}$ and using their directions $\bar{\boldsymbol{r}}_{ij(\boldsymbol{n})}$ to determine axes $\{\boldsymbol{e}_1, \boldsymbol{e}_2, \boldsymbol{e}_3\}$ of $F_i$. Specifically, the first axis, $\boldsymbol{e}_1$, is selected as $\bar{\boldsymbol{r}}_{ij(\boldsymbol{n})}$ with the maximum weight $w_{ij(\boldsymbol{n})}$. For the second axis, we find $\bar{\boldsymbol{r}}_{ij(\boldsymbol{n})}$ with the maximum adjusted-weight $(1 - |\boldsymbol{e}_1 \cdot \boldsymbol{r}_{ij(\boldsymbol{n})}|)w_{ij(\boldsymbol{n})}$, avoiding a direction parallel to $\boldsymbol{e}_1$. The selected vector, denoted as $\bar{\boldsymbol{r}}_2$, is further orthogonalized using the Gram-Schmidt method as $\hat{\boldsymbol{e}}_2 \leftarrow \bar{\boldsymbol{r}}_2 - (\boldsymbol{e}_1 \cdot \bar{\boldsymbol{r}}_2)\boldsymbol{e}_1$, and normalized to a unit vector as $\boldsymbol{e}_2 \leftarrow \hat{\boldsymbol{e}}_2/\|\hat{\boldsymbol{e}}_2\|_2$. Finally, the third axis is simply obtained as $\boldsymbol{e}_3 \leftarrow \boldsymbol{e}_1 \times \boldsymbol{e}_2$, which ensures the orthogonality and $\det(F_i) = 1$. In this process, multiple atoms may have the same weight. For this ambiguity, we add small perturbation noise to each weight $w_{ij(\boldsymbol{n})}$, resulting in randomly selecting a single frame from possible ones. This perturbation scheme is considered a type of stochastic FA (Duval et al., 2023) outlined in Sec. 2.3.

Since these frame construction processes are not stably differentiable, we omit the computation of the gradients from frames $F_i$ to weights $w_{ij(\boldsymbol{n})}$ during training[2]. Still, weights $w_{ij(\boldsymbol{n})}$ receive gradients from $\boldsymbol{x}'$ in Eq. 5 to learn their main function: allowing or blocking messages $\boldsymbol{f}_{i \leftarrow j(\boldsymbol{n})}$ from $j(\boldsymbol{n})$ to $i$. Therefore, we can train a network with dynamic frame construction without using frame gradients.

## 3.2 CRYSTALFRAMER ARCHITECTURE

We demonstrate the proposed concept using Crystalformer (Taniai et al., 2024) as the baseline, as mentioned in Sec. 2.2, and develop a new architecture, CrystalFramer (Fig. 2).

We here regard Eq. 3 as Eq. 5. Thus, we regard the softmax self-attention weights (*i.e.*, exponential weights normalized by $Z_i$ in Eq. 3) as dynamic scaling weights $w_{ij(\boldsymbol{n})}$ in each message-passing layer (Eq. 5). Likewise, we regard the position-augmented value vectors, $\boldsymbol{v}_j + \boldsymbol{\psi}_{ij(\boldsymbol{n})}$, as messages $\boldsymbol{f}_{i \leftarrow j(\boldsymbol{n})}$. During the state update of each atom $\boldsymbol{x}_i$ using Eq. 3, we first compute the attention weights as $w_{ij(\boldsymbol{n})}$. Then, we dynamically construct a local frame for each atom $i$ as matrix $F_i$, by following one of the procedures outlined in Sec. 3.1. Finally, we compute $\boldsymbol{\psi}_{ij(\boldsymbol{n})}$ using $F_i$ and perform Eq. 3. Below, we explain how to derive invariant edge features $\boldsymbol{\psi}_{ij(\boldsymbol{n})}$, given that frame $F_i$ is determined.

**Invariant edge features using a dynamic frame.** For invariant edge feature $\boldsymbol{\psi}_{ij(\boldsymbol{n})}$, Crystalformer originally uses linearly projected Gaussian basis functions (GBFs) to encode distance $r_{ij(\boldsymbol{n})}$. Specifically, GBFs are defined as a mapping from a scalar to a vector of pre-defined dimension $D$, $\boldsymbol{b}(x) = [b_1, b_2, \cdots, b_D]^T$, whose $k$-th component is computed as a Gaussian given by

$$b_k(x; \mu_k, \sigma_k) = \exp\left(-(x - \mu_k)^2/2\sigma_k^2\right). \tag{6}$$

Here, $\mu_k$ and $\sigma_k$ are pre-defined as $\mu_k = \mu_{\min} + (k-1)(\mu_{\max} - \mu_{\min})/(D-1)$ and $\sigma_k = s(\mu_{\max} - \mu_{\min})/(D-1)$, where $\{\mu_{\max}, \mu_{\min}, s, D\}$ are hyperparameters. Intuitively, $\boldsymbol{b}(x)$ encodes scalar $x$ into a soft one-hot vector using $D$ Gaussians uniformly distributed between $\mu_{\min}$ and $\mu_{\max}$. The widths of these Gaussians, $\sigma_k$, are proportional to the interval distance, controlled by scaling factor $s$.

---

[1] We confirmed that covariance matrices $\Sigma_i$ computed with a pretrained Crystalformer model suffered from eigenvalue degeneration at two degrees in about 10% of cases and at three degrees in about 1% of cases. These cases cause rotation ambiguities for two or three (all) axes of $F_i$. To mitigate this issue, we add small perturbation noise to $w_{ij(\boldsymbol{n})}$ in $\Sigma_i$, which stochastically breaks the symmetries in the structural data and empirically helps to compute non-degenerate eigenvalues and eigenvectors. This scheme is considered a type of stochastic FA.

[2] The gradients of the eigenvectors in PCA become numerically unstable when the eigenvalues are degenerate, as the gradients depend on the computation of $1/(\lambda_i - \lambda_j)$ for $i \neq j$. Also, the max-frame procedure is not differentiable due to the use of argmax operations. Although we tried approximating the gradients of argmax, for example, by using a straight-through estimator technique or temperature annealing of softmax, simply ignoring the frame gradients gave the best results.

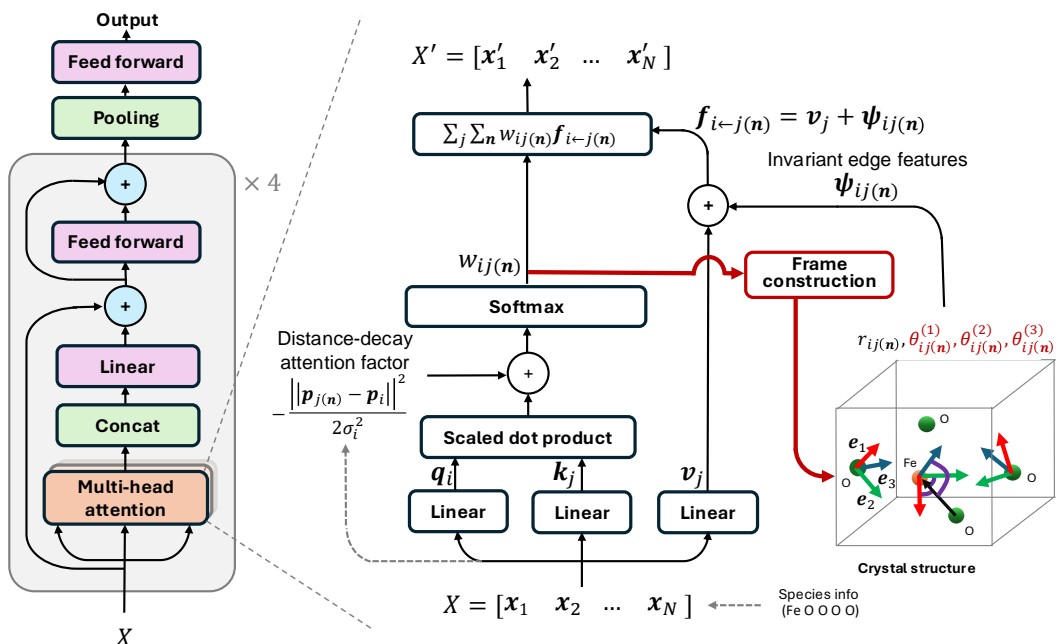

Figure 2: **CrystalFramer architecture.** Dynamic frame construction and frame-based invariant edge features (highlighted in red) are introduced to a transformer for crystals (Taniai et al., 2024).

We retain their distance-based edge feature and further add frame-based edge features to $\boldsymbol{\psi}_{ij(\boldsymbol{n})}$. Specifically, following existing work (Yan et al., 2024), we represent direction vector $\bar{\boldsymbol{r}}_{ij(\boldsymbol{n})}$ invariantly by projecting it onto the frame coordinate system, as $\boldsymbol{\theta}_{ij(\boldsymbol{n})} = F_i \bar{\boldsymbol{r}}_{ij(\boldsymbol{n})}$. Its $k$-th component is calculated as $\boldsymbol{e}_k \cdot \bar{\boldsymbol{r}}_{ij(\boldsymbol{n})}$, the cosine of the angle between $k$-th frame axis $\boldsymbol{e}_k$ and direction $\bar{\boldsymbol{r}}_{ij(\boldsymbol{n})}$. Each component is then converted to a vector using GBFs. By combining the distance-based and three angle-based features via linear projections, we obtain our geometric relative position encoding:

$$\boldsymbol{\psi}_{ij(\boldsymbol{n})} = W_0 \boldsymbol{b}_{\text{dist}}\left(r_{ij(\boldsymbol{n})}\right) + \sum_{k=1,2,3} W_k \boldsymbol{b}_{\text{angl}}\left(\theta_{ij(\boldsymbol{n})}^{(k)}\right). \tag{7}$$

This $\boldsymbol{\psi}_{ij(\boldsymbol{n})}$ as a whole essentially encodes the 3D relative position vector: $\boldsymbol{r}_{ij(\boldsymbol{n})} = \boldsymbol{p}_{j(\boldsymbol{n})} - \boldsymbol{p}_i$. Furthermore, the angle components of $\boldsymbol{\psi}_{ij(\boldsymbol{n})}$ can be interpreted as encoding the deviations of $\boldsymbol{r}_{ij(\boldsymbol{n})}$ from the three primary directions of interatomic interactions (*i.e.*, $\boldsymbol{e}_1, \boldsymbol{e}_2, \boldsymbol{e}_3$) around target atom $i$. Here, four weight matrices $\{W_0, W_1, W_2, W_3\}$ are trainable parameters provided per layer. We also use two types of GBFs, $\boldsymbol{b}_{\text{dist}}$ and $\boldsymbol{b}_{\text{angl}}$, with different hyperparameters for the distance and angles. Specifically, we set $\{\mu_{\min}, \mu_{\max}, s, D\}$ to $\{\frac{14.0}{64}\text{Å}, 14.0\text{Å}, 1.0, 64\}$ for $\boldsymbol{b}_{\text{dist}}$, as suggested by Taniai et al. (2024). We also set them to $\{-1.0, 1.0, 4.0, 64\}$ for $\boldsymbol{b}_{\text{angl}}$, using the cosine range $[-1.0, 1.0]$ and a relatively larger width-scale $s$, which empirically works better for angles. If $\bar{\boldsymbol{r}}_{ij(\boldsymbol{n})}$ is undefined due to division by zero (*i.e.*, when $j(\boldsymbol{n}) = i$), we set $\boldsymbol{b}_{\text{angl}}(\bar{\boldsymbol{r}}_{ij(\boldsymbol{n})}) = \boldsymbol{0}$.

**Overall architecture.** The proposed network precisely follows the Crystalformer architecture (Taniai et al., 2024) as shown in Fig. 2, with the addition of frame construction (Sec. 3.1) and angular edge features (Eq. 7), highlighted in the figure. As we will see in Sec. 5, these simple extensions lead to drastic performance improvements over the baseline. Below, we summarize the key design aspects of the network. The overall architecture consists of an input atom-embedding layer, a stack of four self-attention blocks, global mean pooling, and a final feed-forward network with two linear layers. The self-attention blocks adopt a normalization-free architecture (left part of Fig. 2) by Huang et al. (2020) for enhancing training stability. The infinite summation in self-attention (Eq. 3) is computed convergently and efficiently. Specifically, we adaptively determine the range of unit-cell shifts $\boldsymbol{n}$ to sufficiently cover the neighbor radius of $3.5\sigma_i$, based on dynamic Gaussian tail-length $\sigma_i$. We also employ multi-head self-attention as in the original transformer (Vaswani et al., 2017), with eight heads. Frames are constructed per unit-cell atom, per head, and per layer. For further architectural details, please refer to the original work (Taniai et al., 2024).

## 4 RELATED WORK

The notion of invariant structural modeling encompasses various invariance properties. The most basic one is invariance to the permutation of data-point indices $i$ in structural data (*i.e.*, $A$ and $P$ in Sec. 2.1). It was first addressed by PointNet (Qi et al., 2017) and DeepSets (Zaheer et al., 2017) and is now inherited by GNNs and transformers. The ML community has subsequently focused on invariance under geometric transformations, such as rotations with or without translations (*i.e.*, SO(3)/O(3) or SE(3)/E(3) invariance). In particular, the periodicity of crystals introduces more complex invariance notions, such as periodic SE(3) invariance (Yan et al., 2022), requiring invariance to unit-cell variations within the same crystal structure. These geometric invariance properties have been explored in three primary approaches: 1) invariant features, 2) equivariant features, and 3) frames. We briefly review these approaches, focusing primarily on crystal structures.

**Invariant features.** The most straightforward approach is to rely entirely on inherently invariant geometric quantities, such as the lengths of relative position vectors, throughout a model (Xie & Grossman, 2018; Chen et al., 2019; Ying et al., 2021; Yan et al., 2022; Taniai et al., 2024). However, such distance-based GNNs and transformers have limited expressibility (Pozdnyakov & Ceriotti, 2022). Thus, recent studies have explored more advanced geometric features, such as the angles between triplets in 3-body interactions (Park & Wolverton, 2020; Choudhary & DeCost, 2021; Chen & Ong, 2022), though at the cost of increased computational complexity. More recently, PotNet (Lin et al., 2023) employed the periodic summation of pre-defined interatomic scalar potentials as more physically informed invariant edge features, compared to distances.

**Equivariant features.** Equivariant networks, grounded in group representation theory, form another active research area in 3D structural modeling and include invariant networks as special cases. While we refer readers to recent surveys (Gerken et al., 2023; Duval et al., 2024; Han et al., 2024) for more comprehensive reviews, the initial approach using GNNs for 3D point clouds and atomic systems was proposed by Thomas et al. (2018). Subsequently, this approach has been extended, for example, to introduce improved nonlinearities (Batzner et al., 2022; Brandstetter et al., 2022), attention mechanisms (Fuchs et al., 2020), or greater efficiency (Liao & Smidt, 2023; Liao et al., 2024) in molecular structure modeling. Essentially, these methods use spherical harmonic representations of unit direction vectors $\bar{r}_{ij}$ as rotation-equivariant edge features, which are then equivariantly transformed through specially designed networks. These equivariant features form type-$L$ vectors in a pyramidal shape, where type-0 features encode invariant scalars and type-1 features represent 3D equivariant vectors, such as forces. However, these networks are constrained by limited nonlinearity forms and the increasing computational complexity required to incorporate higher-frequency components in higher degrees $L$. Due to these limitations, their application in crystals is more limited compared to molecules. For example, eComFormer (Yan et al., 2024) has exploited equivariant features in part within each message-passing block for invariant crystal property prediction.

**Frames.** As explained in Sec. 2.3, Puny et al. (2022) introduced FA and applied PCA frames for molecules. Duval et al. (2023) further extended this work in two ways: by proposing stochastic FA to improve efficiency and by applying PCA frames to crystals by treating their unit-cell structures $P$ as finite point clouds. Cheng et al. (2021) used plane waves in crystal structures as invariant positional features, implicitly employing reciprocal lattice vectors as a frame. Similarly, Yan et al. (2024) proposed iComFormer, using transformed lattice vectors with reduced ambiguities as a frame. Lin et al. (2024) proposed minimal FA, an efficient FA method ensuring exact invariance and equivariance.

Our work contributes to this line of research on frame-based invariant networks, providing a new perspective on the previous notion of frames through the introduction of dynamic frames. While several local frame methods exist in the molecular modeling literature (Du et al., 2022; 2023; Pozdnyakov & Ceriotti, 2023), they do not incorporate dynamic frames as we do (see Appendix B for a comparative discussion). We integrate these dynamic frames into a simple distance-based transformer model for crystals (Taniai et al., 2024) to boost its expressive power.

## 5 EXPERIMENTS

To validate the effectiveness of the proposed dynamic frames, we conducted extensive experiments on crystal property prediction. We compared our method with conventional PCA frames (Duval et al., 2023), lattice frames (Yan et al., 2024), and other state-of-the-art architectures for crystals.

**Datasets.** We use three datasets: JARVIS (55,723 materials), MP (69,239 materials), and OQMD (817,636 materials), using snapshots available through a Python package (jarvis-tools). These datasets provide several material properties, such as formation energy and bandgap, simulated by DFT calculations. For further dataset descriptions, see Appendix C. Choudhary & DeCost (2021) and Yan et al. (2022) evaluated many methods on the JARVIS and MP datasets using consistent data splits. Following these and later studies (Lin et al., 2023; Yan et al., 2024; Taniai et al., 2024), we use the same data splits and cite their reported scores to reduce computational burden. Unlike these studies, we also use the much larger-scale OQMD dataset to assess scalability.

**Training settings.** To assess the pure effects of introducing the frames, we precisely follow the training settings of the baseline method, Crystalformer (Taniai et al., 2024), with only one modification. We have increased the number of training epochs to account for the increased complexity of our edge feature design (*i.e.*, our method takes longer to converge, but reduces validation losses more rapidly). Specifically, for the JARVIS dataset, we train our model from scratch by optimizing the mean absolute loss function using Adam (Kingma & Ba, 2015) for a total of 2000 epochs, while enabling the frames from the beginning. A summary of detailed training settings, including the number of epochs, batch size, and learning rate for the three datasets, can be found in Appendix D.

## 5.1 CRYSTAL PROPERTY PREDICTION

Tables 1 and 2 extensively compare the mean absolute errors of the proposed and existing methods for the JARVIS (5 tasks) and MP (4 tasks) datasets. Several earlier methods are omitted from the tables and provided in Appendix E. Overall, our method with max frames outperforms others in most tasks, significantly enhancing the performance of the baseline Crystalformer model. Such improvements never fade even when feeding the much larger OQMD dataset, as shown in Table 3. It is important to note that the current state-of-the-art, ComFormer, uses finely-tuned hyperparameters (*e.g.*, learning rate, loss function, number of layers, graph structure) for each individual task, whereas we simply adjust the number of epochs and batch size for each dataset. In the bottom parts of Tables 1 and 2, our weighted PCA frame method shows relatively limited improvements (see Appendix F for a detailed discussion). Nevertheless, it outperforms its conventional counterpart using PCA frames. Additionally, we evaluated a variant using static local frames. These frames are similar to max frames but constructed with static weights, $w_{ij(\boldsymbol{n})} = \exp\left(-r_{ij(\boldsymbol{n})}^2\right)$. As a result, these static local frames rely only on the distances to neighbors and do not account for dynamic self-attention weights. The max frame method outperforms this static counterpart in most tasks. These results successfully validate the effectiveness of our concept of dynamic frames.

## 5.2 EFFICIENCY COMPARISON

Table 4 compares the model efficiencies of several top-performing architectures. Notably, despite the superior performance of the proposed method, it requires significantly fewer parameters than PotNet, Matformer, and iComFormer. Compared to Crystalformer, our method introduces only a small overhead of about 100K parameters owing to projection matrices $\{W_1, W_2, W_3\}$ in Eq. 7. Given the performance gains shown in Tables 1–3, this high cost-performance ratio also highlights the effectiveness of our dynamic frame feature design. In terms of runtime, the test time is faster than PotNet, Matformer, and iComFormer, which are hindered by relatively heavy data preprocessing. However, compared to Crystalformer, the training and test times are more than double, mainly due to the increased computation cost of $\sum_{\boldsymbol{n}} w_{ij(\boldsymbol{n})} \psi_{ij(\boldsymbol{n})}$. As noted by Taniai et al. (2024), this term is the primary bottleneck in Crystalformer and also in our model. In Appendix G (Tables A2–A5), we show that a lightweight configuration for GBFs reduces the training time by approximately 42% while maintaining comparable accuracy. Pre-training without frames to efficiently learn attention weights first may also accelerate training. In Appendix H, we further discuss scalability for large structures.

## 6 DISCUSSION AND LIMITATIONS

**Visual analysis.** Figure 3 displays four types of frames generated for a test material (JVASP-30609) in the JARVIS formation energy prediction task. While PCA (Duval et al., 2023) and lattice (Yan et al., 2024) frames are static, the proposed weighted PCA and max frames exhibit dynamic behavior based on learned attention weights. In each layer, our frames capture distinct local motifs, such as

Table 1: **Property prediction results on the JARVIS dataset.** Accuracies are in mean absolute error. The sizes of training, validation, and test splits are listed under each property name. **Bold** indicates the best results, underline the second best. Full results covering earlier methods are in Appendix E.

| Method | Form. energy 44578 / 5572 / 5572 eV/atom | Total energy 44578 / 5572 / 5572 eV/atom | Bandgap (OPT) 44578 / 5572 / 5572 eV | Bandgap (MBJ) 14537 / 1817 / 1817 eV | E hull 44296 / 5537 / 5537 eV |
|---|---|---|---|---|---|
| Matformer (Yan et al., 2022) | 0.0325 | 0.035 | 0.137 | 0.30 | 0.064 |
| PotNet (Lin et al., 2023) | 0.0294 | 0.032 | 0.127 | 0.27 | 0.055 |
| eComFormer (Yan et al., 2024) | 0.0284 | 0.032 | 0.124 | 0.28 | **0.044** |
| iComFormer (Yan et al., 2024) | 0.0272 | 0.0288 | 0.122 | 0.26 | 0.047 |
| Crystalformer (Taniai et al., 2024) | 0.0306 | 0.0320 | 0.128 | 0.274 | 0.0463 |
| — w/ PCA frames (Duval et al., 2023) | 0.0325 | 0.0334 | 0.144 | 0.292 | 0.0568 |
| — w/ lattice frames (Yan et al., 2024) | 0.0302 | 0.0323 | 0.125 | 0.274 | 0.0531 |
| — w/ static local frames | 0.0285 | 0.0292 | 0.122 | 0.261 | **0.0444** |
| — w/ weighted PCA frames (proposed) | 0.0287 | 0.0305 | 0.126 | 0.279 | **0.0444** |
| — w/ max frames (proposed) | **0.0263** | **0.0279** | **0.117** | **0.242** | 0.0471 |

Table 2: **Property prediction results on the MP dataset.**

| Method | Formation energy 60000 / 5000 / 4239 eV/atom | Bandgap 60000 / 5000 / 4239 eV | Bulk modulus 4664 / 393 / 393 log(GPa) | Shear modulus 4664 / 392 / 393 log(GPa) |
|---|---|---|---|---|
| Matformer | 0.021 | 0.211 | 0.043 | 0.073 |
| PotNet | 0.0188 | 0.204 | 0.040 | 0.065 |
| eComFormer | 0.0182 | 0.202 | 0.0417 | 0.0729 |
| iComFormer | 0.0183 | 0.193 | 0.0380 | **0.0637** |
| Crystalformer | 0.0186 | 0.198 | 0.0377 | 0.0689 |
| — w/ PCA frames | 0.0197 | 0.217 | 0.0424 | 0.0719 |
| — w/ lattice frames | 0.0194 | 0.212 | 0.0389 | 0.0720 |
| — w/ static local frames | 0.0178 | 0.191 | 0.0354 | 0.0708 |
| — w/ weighted PCA frames (proposed) | 0.0197 | 0.214 | 0.0423 | 0.0715 |
| — w/ max frames (proposed) | **0.0172** | **0.185** | **0.0338** | 0.0677 |

Table 3: **Property prediction results on the OQMD dataset.**

| Method | Form. energy (eV/atom) 654108 / 81763 / 81763 | Bandgap (eV) 653388 / 81673 / 81673 | E hull (eV/atom) 654108 / 81763 / 81763 |
|---|---|---|---|
| Crystalformer | 0.02115 | 0.06028 | 0.06759 |
| **CrystalFramer** (max frames) | **0.01871** | **0.05805** | **0.06607** |

Table 4: **Efficiency comparison.** Per-epoch training time includes validation, and per-material test time includes preprocessing, such as graph construction. The runtimes are evaluated for the formation energy prediction in the JARVIS dataset using a single NVIDIA A6000 GPU with 48GB VRAM.

| Model | Arch. type | Time/epoch | Test/mater. | #Params. | #Params./block |
|---|---|---|---|---|---|
| PotNet | GNN | 43 s | 313 ms | 1.8 M | 527 K |
| Matformer | Transformer | 60 s | 20.4 ms | 2.9 M | 544 K |
| iComFormer | Transformer | 59 s | 54.8 ms | 5.0 M | 855 K |
| Crystalformer | Transformer | 32 s | 6.6 ms | 853 K | 206 K |
| **CrystalFramer** | Transformer | 74 s | 16.8 ms | 952 K | 231 K |

octahedra with a green central magnesium atom surrounded by blue fluorine atoms, and tetrahedra with a red central tin atom surrounded by blue magnesium atoms. These local structures are common and often distorted, as seen in this example. The ability to capture these local structures and measure distortions via relative positions may contribute to the high performance observed. A more detailed analysis, including comparative discussions on different frames, visualizations for a different material, and an examination of frame evolution during training, is provided in Appendix F. In particular, the frame evolution analysis (Appendix F.3) shows that max frames converge faster during training due to the discrete nature of their construction. While this characteristic may contribute to the superior performance of max frames, it introduces noticeable discontinuities to the model and may limit generalization to out-of-domain data, as discussed in Appendix I.

**Baseline choice.** This study adopts Crystalformer (Taniai et al., 2024) for demonstration, since its standard multi-head softmax attention is well-suited for dynamic frames. Other crystal transformers (Yan et al., 2022; 2024) use distinct channel-wise sigmoid attention, akin to maximally

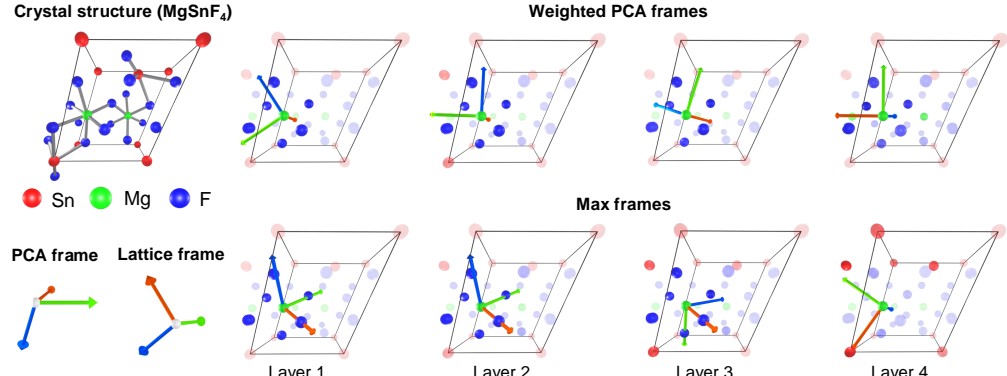

Figure 3: **Frame visualizations.** Conventional PCA and lattice frames provide a global coordinate system based solely on the structure. The proposed dynamic frames extract different structural information for each atom and layer using dynamic attention weights, shown as varying transparency.

multi-headed attention. Since frames and angular features are computed per atom, per head, and per layer, such channel-wise attention is not preferable. On the other hand, Crystalformer with Eq. 3 can be interpreted as the original fully-connected self-attention (Vaswani et al., 2017), $\boldsymbol{x}_i' = Z_i^{-1} \sum_j \exp(\boldsymbol{q}_i^T \boldsymbol{v}_j / \sqrt{d_K} + \phi_{ij})(\boldsymbol{v}_j + \boldsymbol{\psi}_{ij})$, with two straightforward extensions: 1) relative position representations via $\phi_{ij}$ and $\boldsymbol{\psi}_{ij}$ (Shaw et al., 2018) and 2) duplication of each atom $j$ as $j(\boldsymbol{n})$ using $\sum_{\boldsymbol{n}}$ for crystal periodicity. Given the practicability and versatility of the original transformer architecture across diverse domains (Lin et al., 2022), our demonstration serves as a foundation for transformer-based crystal encoders using dynamic frames.

**Equivariant prediction.** While this study focuses on SE(3) invariance, Puny et al. (2022) extended FA to predict equivariant quantities, such as force vectors, by applying inverse mapping $F^{-1}$ to $f(X, FP)$ in Eq. 4 prior to averaging. In our case, one potential equivariant extension would thus invariantly output atom-wise geometric quantities $\boldsymbol{u}_i$ from $\boldsymbol{x}_i'$ (*e.g.*, via $\boldsymbol{u}_i = W\boldsymbol{x}_i'$) and subsequently apply the inverse mapping as $F_i^{-1}\boldsymbol{u}_i$. Another approach, similar to recent work (Shi et al., 2023), could tie the outputs equivariantly to the input structure, for instance, by computing $\boldsymbol{u}_i = \sum_j \sum_{\boldsymbol{n}} w_{ij(\boldsymbol{n})} \boldsymbol{r}_{ij(\boldsymbol{n})}$. These equivariant extensions would enable predictions of forces and relaxed structures, which are crucial for surface property analysis (Chanussot et al., 2021; Tran et al., 2023; Bihani et al., 2024). Detailed investigations into these equivariant extensions are left for future work.

**Application to molecules.** Transformers for molecular structures have been developed (Ying et al., 2021; Wang et al., 2023; Shi et al., 2023; Liu et al., 2024), and our dynamic frames could potentially be applied to them. However, crystal and molecular structures have very different characteristics. In particular, molecular structures often contain so few atoms that they tend to form low-dimensional local structures, which may hinder the construction of effective frames. Extending our methodology to molecules represents another intriguing future direction for this research.

## 7 CONCLUSIONS

This study revisited the challenge of determining effective frames for the SE(3)-invariant modeling of crystal structures. We hypothesized that frames should reflect the local dynamic environment around each atom, rather than the static global structure. We herein introduced the concept of *dynamic frames*, which leverage the strengths of interatomic interactions. These frames were integrated into an existing transformer-based network for crystal property prediction (Taniai et al., 2024), resulting in the CrystalFramer architecture. Its performance was benchmarked against conventional frame construction methods (Duval et al., 2023; Yan et al., 2024) and other state-of-the-art networks (Choudhary & DeCost, 2021; Yan et al., 2022; Lin et al., 2023). Our findings validated the hypothesis, demonstrating the superior performance of the proposed approach. Although the demonstration focused on crystals, interaction-based dynamic frame construction holds promise for broader applications, including surface modeling, molecular structure modeling, and ML-driven simulations of particles and fluids.

AUTHOR CONTRIBUTIONS

**YI**, as an intern at OMRON SINIC X, identified the potential utility of the proposed idea for crystals, implemented most of the method in the baseline code, designed and conducted the majority of the experiments, analyzed the results, drafted the initial manuscript, and created the figures. **TT** conceived the method, developed and discussed the core idea and philosophy with the co-authors, co-patented the method with YI and RI, explained the baseline code to YI, helped with the coding and experiments, revised the manuscript draft, and led the rebuttal and overall project. **RI** provided expertise in materials science and physics simulations, helped review the molecular modeling literature, and advised on the methodology, coding, and writing. **YU** co-led materials-related collaborations, provided expertise in machine learning, and advised on the methodology and writing. **KO** co-led materials-related collaborations, provided expertise in materials science, and advised on the writing.

ACKNOWLEDGMENTS

This work was supported in part by JST-Mirai Program Grant Number JPMJMI21G2 and JST Moonshot R&D Program Grant Number JPMJMS2236. TT is supported in part by JSPS KAKENHI Grant Number 24K23911. RI is supported in part by JSPS KAKENHI Grant Number 24K23910. YU is partly supported by JST-Mirai Program Grant Number JPMJMI21G2 and JST Moonshot R&D Program Grant Number JPMJMS2236. TT thanks Naoya Chiba for his insightful comment on the weakness of PCA for symmetries, which provided an initial hint for this work. The authors also thank Yuta Suzuki, Seiji Aota, and the anonymous reviewers for their valuable feedback on the manuscript.

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

## A  LIMITATIONS OF UNIT-CELL-BASED CRYSTAL REPRESENTATIONS

The conventional PCA frames explained in Sec. 2.3 implicitly assume a unique lattice representation such as the Niggli reduced cell (Niggli, 1928). Similarly, the lattice frames assume a primitive cell and convert it to a cell similar to the reduced one. Otherwise, these frames are affected by the arbitrariness of unit cell representations, such as supercells and conventional cells.

Traditionally, primitive cells and conventional cells are used to represent periodic structures. Primitive cells are defined as the smallest repeating units of a lattice, having the minimum volume and containing only a single lattice point within each cell. By following a mathematical procedure on primitive cells, their unique representations called reduced unit cells can be obtained (Santoro & Mighell, 1970).

On the other hand, conventional cells are defined as unit cells that are not necessarily primitive but are designed to exhibit symmetry in an easily understandable way. The notion of conventional cells is often illustrated by the face-centered cubic lattice and the body-centered cubic lattice.

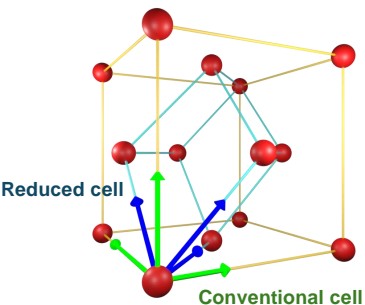

Figure A1: **Conventional cell (green) and Niggli reduced cell (blue) for a face-centered cube.**

Figure A1 compares a conventional cell and the Niggli reduced cell of a face-centered cubic structure. Examining the conventional unit cell easily reveals that it represents a cubic lattice, with atoms located at each corner and at each face center. However, this fact is obscured in the reduced cell. Therefore, reduced cells can be said to sacrifice the interpretability of physically important information, such as symmetry, in order to uniquely represent periodic structures.

## B    COMPARISON TO EXISTING LOCAL FRAMES FOR MOLECULES

In the molecular modeling literature, several local frames have been proposed (Du et al., 2022; 2023; Pozdnyakov & Ceriotti, 2023). The concept of our dynamic frames, being both dynamic and local, is distinct from these frames for molecules, which are local but static. Below we discuss this perspective in more detail.

We first clarify the terminology regarding 'dynamic' and 'static' in this context. We use 'dynamic' to describe behavior that is influenced by the model's internal states estimated for a given structure. For instance, interatomic interactions modeled within a GNN reflect these internal states and evolve dynamically layer by layer. Dynamic frames are designed to align with these interatomic interactions. While the molecular modeling literature often uses 'dynamic' to describe temporally evolving structures, our work does not assume such temporal dynamics. Similarly, we use 'static' to describe behavior that is unaffected by the model's internal states.

**Du et al. (2022) propose static edge-wise frames.** These edge-wise frames, denoted as $F_{ij} = [e_1, e_2, e_3]^T$ using our notation, are $3 \times 3$ orthogonal matrices defined individually for each edge $(i, j)$. From Eq. 2 in their paper, the axes of $F_{ij}$ are defined as $e_1 = \text{unit}(p_i - p_j)$, $e_2 = \text{unit}(p_i \times p_j)$, and $e_3 = e_1 \times e_2$, where $\text{unit}(x) = x/\|x\|$ is $L_2$ normalization. Here, the centroid of the structure is pre-shifted to the origin, as $p \leftarrow p - \bar{p}$ using $\bar{p} = \frac{1}{N} \sum_i p_i$. Thus, these frames are translation invariant, even though $e_2$ appears to depend on absolute positions. However, performing such a global centroid shift for crystals is not straightforward due to their infinite periodicity, unless a specific unit cell description is utilized.

**Du et al. (2023) propose frame-based equivariant message passing using static edge-wise and node-wise frames.** These edge-wise frames $F_{ij}$ are identical to those used in their earlier work (Du et al., 2022) (see above). Their node-wise frames $F_i$ are defined similarly to $F_{ij}$, but with $p_j$ replaced by the cluster centroid around $i$: $\bar{p}_i = \frac{1}{|N(i)|} \sum_{j \in N(i)} p_j$. Thus, the axes of $F_i$ are provided as $e_1 = \text{unit}(p_i - \bar{p}_i)$, $e_2 = \text{unit}(p_i \times \bar{p}_i)$, and $e_3 = e_1 \times e_2$. (See Eqs. 13 and 14 in their paper for the definitions.) To ensure translation invariance, these node-wise frames also rely on global centroid normalization. Moreover, when applied to crystal structures. their highly symmetric nature will often cause $\bar{p}_i \simeq p_i$, resulting in unstable frame construction.

**Pozdnyakov & Ceriotti (2023) propose ensemble of many 3-body interactions** called the equivariant coordinate-system ensemble. For each target atom $i$, they construct many triplets of atoms $(i, j, j')$ using pairs of neighbors $(j, j')$ and then construct a local frame for each triplet as $F_{ijj'}$. Although these triplet-wise frames are local, they do not reflect dynamic internal states of the model. Also, modeling 3-body interactions is computationally expensive.

Overall, these methods all employ specific types of static local frames, such as node-wise (Du et al., 2023), edge-wise (Du et al., 2022; 2023), or triplet-wise (Pozdnyakov & Ceriotti, 2023) frames. None of them leverage the model's internal states for frame construction.

In Sec. 5.1, we further compare the proposed method using dynamic frames with its static counterpart variant, which is based on static local frames. The results in Tables 1 and 2 demonstrate the superior performance of the proposed dynamic frames, highlighting the conceptual difference between these two families of frames.

## C    DATASET SPECIFICATIONS

We use the following three sources of materials data for evaluations. They are all publicly available through a Python package (jarvis-tools) created by Choudhary et al. (2020).

**The JARVIS-DFT 3D 2021** is a collection of 55,723 materials provided by Choudhary et al. (2020) and is accessible as `dft_3d_2021` via jarvis-tools (or as `dft_3d` in older versions). These materials are annotated with various simulated properties using two DFT calculation methods, OptB88vdW (OPT) and TBmBJ (MBJ). Following recent studies (Yan et al., 2022; 2024; Lin et al., 2023; Taniai et al., 2024), we use formation energy (`formation_energy_peratom`), total energy (`optb88vdw_total_energy`), bandgap (`optb88vdw_bandgap` and `mbj_bandgap`), and energy above hull or E hull (`ehull`) as regression targets.

**The Materials Project (MP) database** (Jain et al., 2013) is an online public materials database providing various synthetic materials and their DFT-calculated properties. We specifically use its snapshot collected by Chen et al. (2019), which contains 69,239 materials and is accessible as `megnet` via jarvis-tools. Following recent studies (Yan et al., 2022; 2024; Lin et al., 2023; Taniai et al., 2024), we use formation energy (`e_form`), bandgap (`gap pbe`), bulk modulus (`bulk modulus`), and shear modulus (`shear modulus`) as regression targets. For bulk and shear modulus, we use the data splits provided by Yan et al. (2022).

**The Open Quantum Materials Database (OQMD)** is another online public materials database by Kirklin et al. (2015). We specifically use its snapshot provided as `oqmd_3d_no_cfid` in jarvis-tools, which contains 817,636 materials with three DFT-calculated properties: formation energy (`_oqmd_delta_e`), bandgap (`_oqmd_band_gap`), and energy above hull (`_oqmd_stability`). We use these properties as regression targets. We release our data splits along with our code.

## D  TRAINING SETTINGS

Table A1 summarizes the training settings for the JARVIS, MP, and OQMD datasets.

Specifically, for the JARVIS dataset, we optimize the mean absolute loss function using the Adam optimizer (Kingma & Ba, 2015) with $(\beta_1, \beta_2) = (0.9, 0.98)$ and weight decay of $10^{-5}$ (Loshchilov & Hutter, 2019). We employ the warm-up-free inverse square root scheduling (Huang et al., 2020) for the learning rate, with the initial learning rate of $5.0 \times 10^{-4}$ and decay factor of $\sqrt{4000/(4000 + t)}$ according to the total train steps $t$. The model weights are initialized through the strategy for the normalization-free transformer architecture by Huang et al. (2020), which improves the training stability. The training is iterated for a total of 2000 epochs with a batch size of 256 materials. Stochastic weight averaging (SWA) (Izmailov et al., 2018) is adopted for model selection for testing and validation. Except for the increased number of epochs, we use the same settings with the baseline Crystalformer model (Taniai et al., 2024) to evaluate the pure effects of introducing the frames.

For the OQMD dataset, which was not used by the baseline method, we use similar settings with a larger batch size of 1024 materials and fewer epochs of 200.

Table A1: **Detailed training settings.**

| Hyperparameters | Settings (JARVIS, MP, OQMD) |
|---|---|
| Loss function | Mean absolute error |
| Optimizer | AdamW with $(\beta_1, \beta_2) = (0.9, 0.98)$ |
| Weight decay | $10^{-5}$ |
| Gradient norm clipping | 1.0 |
| Initial learning rate $\alpha$ | $5.0 \times 10^{-4}$ |
| Learning rate scheduling per step | $\alpha\sqrt{4000/(4000 + t)}$ |
| Warm-up steps | 0 (no warm-up) |
| Batch size | 256, 128, 1024 |
| Number of epochs | 2000, 800, 200 |
| Dropout rate | 0.0 |
| SWA epochs | 50, 50, 20 |

## E  FULL BENCHMARK RESULTS

Tables A2 and A3 provide the full versions of Tables 1 and 2, adding the results of CGCNN (Xie & Grossman, 2018), SchNet (Schütt et al., 2018), MEGNet (Chen et al., 2019), GATGNN (Louis et al.,

2020), M3GNet (Chen & Ong, 2022), and ALIGNN (Choudhary & DeCost, 2021). These tables also include the comparisons between the default and lightweight configurations of CrystalFramer described in Appendix G.

We also report additional results for the OQMD dataset using larger models in Table A4. Increasing the number of self-attention blocks enhances the performances of both the baseline and proposed models. In both the four and eight block models, the version with max frames consistently outperforms the baseline. Investigating even larger models is left for future work.

## F    DETAILED VISUAL ANALYSIS OF FRAMES

### F.1    COMPARISON BETWEEN WEIGHTED PCA FRAMES AND MAX FRAMES

As shown in Tables 1 and 2, the max frame method performed very well, while the weighted PCA variant did not. In Fig. 3, the weighted PCA frames do not seem to capture the local structure very well compared to the max frames. This is because all the attention weights, even small ones, can influence the composition of the weighted PCA frames. In other words, the weighted PCA frames look at the structure over a broader area, while the max frames focus on relatively close neighbors. This difference seems to have a positive effect on the max frames and a negative effect on the weighted PCA frames in most tasks, except for the E hull in the MP dataset (Table 2).

For the E hull prediction, it is suggested by Taniai et al. (2024) that the inclusion of long-range interatomic interactions is a critical factor. This implication can reasonably explain the better performance of the weighted PCA frames for the E hull. That is, the weighted PCA frames emphasize distant atoms and help deliver more meaningful messages from these distant atoms that are important for the E hull prediction.

### F.2    FRAME VISUALIZATIONS FOR A DIFFERENT MATERIAL

Figure A2 shows the frame visualizations for another test material (JVASP-85272). This structure consists of carbon (red atoms) and nitrogen (blue atoms), forming a tetrahedral structure. Both dynamic models first attend to the central tetrahedral structure in the first two layers, and then increase the attention to relatively distant red atoms in the subsequent layers. However, the max frames capture these structures more clearly than the weighted PCA frames, as observed in the first example.

We have also noticed a general tendency for our models to attend to close neighbors in shallow layers and relatively distant neighbors in deeper layers. This tendency is also reasonable. Since the states of atoms are initialized as symbolic atomic species without rich information, they must gather information about their surroundings in shallow layers to configure their states. In deeper layers, these atoms become ready to engage in complex interactions with selected distant atoms.

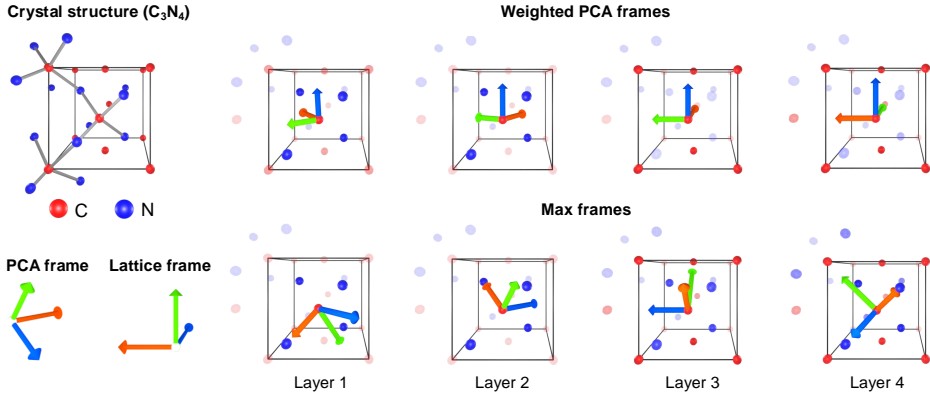

Figure A2: **Frame visualizations for a different material.**

Table A2: **Property prediction results on the JARVIS dataset (full).**

| Method | Form. energy 44578 / 5572 / 5572 eV/atom | Total energy 44578 / 5572 / 5572 eV/atom | Bandgap (OPT) 44578 / 5572 / 5572 eV | Bandgap (MBJ) 14537 / 1817 / 1817 eV | E hull 44296 / 5537 / 5537 eV |
|---|---|---|---|---|---|
| CGCNN (Xie & Grossman, 2018) | 0.063 | 0.078 | 0.20 | 0.41 | 0.17 |
| SchNet (Schütt et al., 2018) | 0.045 | 0.047 | 0.19 | 0.43 | 0.14 |
| MEGNet (Chen et al., 2019) | 0.047 | 0.058 | 0.145 | 0.34 | 0.084 |
| GATGNN (Louis et al., 2020) | 0.047 | 0.056 | 0.17 | 0.51 | 0.12 |
| M3GNet (Chen & Ong, 2022) | 0.039 | 0.041 | 0.145 | 0.362 | 0.095 |
| ALIGNN (Choudhary & DeCost, 2021) | 0.0331 | 0.037 | 0.142 | 0.31 | 0.076 |
| Matformer (Yan et al., 2022) | 0.0325 | 0.035 | 0.137 | 0.30 | 0.064 |
| PotNet (Lin et al., 2023) | 0.0294 | 0.032 | 0.127 | 0.27 | 0.055 |
| eComFormer (Yan et al., 2024) | 0.0284 | 0.032 | 0.124 | 0.28 | **0.044** |
| iComFormer (Yan et al., 2024) | 0.0272 | 0.0288 | 0.122 | 0.26 | 0.047 |
| Crystalformer (Taniai et al., 2024) | 0.0306 | 0.0320 | 0.128 | 0.274 | 0.0463 |
| — w/ PCA frames (Duval et al., 2023) | 0.0325 | 0.0334 | 0.144 | 0.292 | 0.0568 |
| — w/ lattice frames (Yan et al., 2024) | 0.0302 | 0.0323 | 0.125 | 0.274 | 0.0531 |
| — w/ static local frames | 0.0285 | 0.0292 | 0.122 | 0.261 | **0.0444** |
| — w/ weighted PCA frames (proposed) | 0.0287 | 0.0305 | 0.126 | 0.279 | **0.0444** |
| — w/ max frames (proposed) | **0.0263** | **0.0279** | **0.117** | **0.242** | 0.0471 |
| CrystalFramer (default) | **0.0263** | **0.0279** | **0.117** | **0.242** | 0.0471 |
| CrystalFramer (lightweight) | 0.0268 | **0.0279** | **0.117** | 0.262 | **0.0467** |

Table A3: **Property prediction results on the MP dataset (full).**

| Method | Formation energy 60000 / 5000 / 4239 eV/atom | Bandgap 60000 / 5000 / 4239 eV | Bulk modulus 4664 / 393 / 393 log(GPa) | Shear modulus 4664 / 392 / 393 log(GPa) |
|---|---|---|---|---|
| CGCNN | 0.031 | 0.292 | 0.047 | 0.077 |
| SchNet | 0.033 | 0.345 | 0.066 | 0.099 |
| MEGNet | 0.030 | 0.307 | 0.060 | 0.099 |
| GATGNN | 0.033 | 0.280 | 0.045 | 0.075 |
| M3GNet | 0.024 | 0.247 | 0.050 | 0.087 |
| ALIGNN | 0.022 | 0.218 | 0.051 | 0.078 |
| Matformer | 0.021 | 0.211 | 0.043 | 0.073 |
| PotNet | 0.0188 | 0.204 | 0.040 | 0.065 |
| eComFormer | 0.0182 | 0.202 | 0.0417 | 0.0729 |
| iComFormer | 0.0183 | 0.193 | 0.0380 | **0.0637** |
| Crystalformer | 0.0186 | 0.198 | 0.0377 | 0.0689 |
| — w/ PCA frames | 0.0197 | 0.217 | 0.0424 | 0.0719 |
| — w/ lattice frames | 0.0194 | 0.212 | 0.0389 | 0.0720 |
| — w/ static local frames | 0.0178 | 0.191 | 0.0354 | 0.0708 |
| — w/ weighted PCA frames (proposed) | 0.0197 | 0.214 | 0.0423 | 0.0715 |
| — w/ max frames (proposed) | **0.0172** | **0.185** | **0.0338** | 0.0677 |
| CrystalFramer (default) | **0.0172** | **0.185** | **0.0338** | 0.0677 |
| CrystalFramer (lightweight) | 0.0176 | 0.191 | 0.0341 | **0.0654** |

Table A4: **Property prediction results on the OQMD dataset (full).**

| Method | # Blocks | Form. energy (eV/atom) 654108 / 81763 / 81763 | Bandgap (eV) 653388 / 81673 / 81673 | E hull (eV/atom) 654108 / 81763 / 81763 |
|---|---|---|---|---|
| Crystalformer (baseline) | 4 | 0.02115 | 0.06028 | 0.06759 |
| **CrystalFramer** (default) | 4 | 0.01871 | 0.05805 | **0.06607** |
| **CrystalFramer** (lightweight) | 4 | **0.01813** | **0.05773** | 0.06672 |
| Crystalformer (baseline) | 8 | 0.02104 | 0.05986 | 0.06690 |
| **CrystalFramer** (default) | 8 | 0.01778 | 0.05785 | 0.06454 |
| **CrystalFramer** (lightweight) | 8 | **0.01731** | **0.05142** | **0.06403** |

Table A5: **Efficiency comparison with the lightweight configuration.** See Appendix G for details.

| Config. | Time/epoch | Test/mater. | #Params. | #Params./block |
|---|---|---|---|---|
| Default | 74 s | 16.8 ms | 952 K | 231 K |
| Lightweight | 43 s | 15.2 ms | 878 K | 212 K |

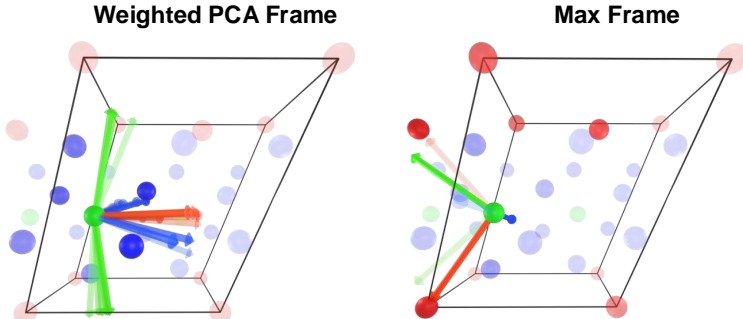

**Weighted PCA Frame**   **Max Frame**

Figure A3: **Evolution of dynamic frames during training.** We visualize the weighted PCA frames and max frames using model checkpoints taken every 200 epochs, starting from epoch 100 until 2000. Frames from earlier checkpoints are overlaid with higher transparency. Notably, the max frames stabilize more quickly than the weighted PCA frames.

### F.3 EVOLUTION OF DYNAMIC FRAMES DURING TRAINING

We further examined how dynamic frames evolve throughout the training process, by visualizing frames using model checkpoints taken at 200-epoch intervals. Figure A3 compares the evolution of the weighted PCA frames and max frames for the same material as Fig. 3. We observed that the weighted PCA frames fluctuated throughout training, whereas the max frames stabilized quickly during the early stages. As frame fluctuations can introduce noise, the early stabilization of the max frames may explain their superior performance compared to the weighted PCA frames.

### G EFFICIENT MODEL CONFIGURATION

As discussed in Sec. 5.2, the key to improving the runtime of our method is to accelerate the infinite summation of edge features: $\sum_{\boldsymbol{n}} w_{ij(\boldsymbol{n})}\boldsymbol{\psi}_{ij(\boldsymbol{n})}$. These edge features $\boldsymbol{\psi}_{ij(\boldsymbol{n})}$ are the sum of one distance-based feature and three angle-based edge features represented via GBFs $\boldsymbol{b}$:

$$\boldsymbol{\psi}_{ij(\boldsymbol{n})} = \lambda \left( c_{\text{dist}} W_0 \boldsymbol{b}_{\text{dist}} \left( r_{ij(\boldsymbol{n})} \right) + c_{\text{angl}} \sum_{k=1,2,3} W_k \boldsymbol{b}_{\text{angl}} \left( \theta_{ij(\boldsymbol{n})}^{(k)} \right) \right). \tag{A1}$$

Here, new coefficient parameters $\lambda, c_{\text{dist}}, c_{\text{angl}} \in \mathbb{R}_+$ have been introduced into the original form in Eq. 7. The default model configuration uses largely overlapping Gaussian bases ($s = 4.0$) for angular features. This means that $\boldsymbol{b}_{\text{angl}}(x)$ changes very smoothly along $x$ and can be well-approximated with fewer Gaussian bases (*i.e.*, lower $D$).

When reducing $D$ to $D'$, one can obtain comparable Gaussian widths and value ranges by converting their scale to $s' = sD'/D$ and coefficient to $c' = cD/D'$. Using this rule, we obtained a lightweight version of the angular feature term by changing $\{D, s, c\}$ from $\{64, 4.0, 1.0\}$ to $\{16, 1.0, 4.0\}$. After hyperparameter tuning, we further increased $\lambda$ from 1.0 to 1.5.

Table A5 compares the efficiencies of CrystalFramer models with the default and lightweight configurations, and Tables A2–A4 compare their performances on the JARVIS, MP, and OQMD datasets. These results show that the lightweight model reduces the train time by 42% while achieving comparable accuracies on the JARVIS and MP datasets and better accuracies on the OQMD dataset.

### H SCALABILITY FOR LARGE STRUCTURES AND SUPERCELLS

Since the proposed CrystalFramer is based on a self-attention mechanism, its computational complexity is $O(Nk)$, where $N$ is the number of atoms in the unit cell and $k$ is the number of neighbors per unit-cell atom. In the infinitely connected attention of Crystalformer (Taniai et al., 2024) defined in

Eq. 3, neighbors $j(\boldsymbol{n})$ are adaptively determined for each atom $i$ in each layer. The current implementation computes neighbors by periodically repeating the unit cell within a finite range. Consequently, $k$ becomes a multiple of $N$, resulting in an overall computational complexity of $O(N^2)$.

In practice, the training of CrystalFramer has successfully scaled to relatively large structures in the MP dataset, which features an average of 30 atoms per unit cell and a maximum of 296 atoms. For inference, the method can handle even larger structures than during training, as it requires significantly less memory and supports per-material (non-batched) processing.

Scalability for larger structures becomes crucial especially when processing supercells. Supercells are often utilized when structures deviate from perfect periodicity, such as in the presence of impurities, defects, or surfaces. We consider the following two potential approaches to improve efficiency with large supercells.

**Mixed atom embedding.** Structures with impurities or defects are often represented using site occupancy, which indicates the probabilities of different elements occupying an atomic site. Instead of modeling such structures with supercells, we can efficiently represent the site occupancy by mixing atomic embedding vectors. In this case, each $a_i$ represents a probability distribution over elements rather than a single element. The corresponding atomic state can then be initialized as a linear blend of atom embeddings: $\boldsymbol{x}_i \leftarrow \sum_{\text{element}} a_i(\text{element})\text{AtomEmbedding}(\text{element})$. This approach can keep the structure size small without using a supercell, thereby maintaining overall efficiency.

**Distance-based neighbor search.** When unit cells are large, the current cell-based neighbor identification method will produce redundant neighbors, forcing $k \geq N$. By employing a more compact set of neighbors through nearest neighbor search, the complexity is reduced from $O(N^2)$ to $O(Nk)$, improving efficiency for larger structures.

Since structures with imperfect periodicity are common in realistic scenarios, developing scalable models for these structures is an important direction for future research.

## I   ANALYSIS OF MODEL'S CONTINUITY

Dym et al. (2024) pointed out that frame-based models generally exhibit discontinuous characteristics, which are also inherent in our approach. To empirically assess the degree of this discontinuity in our trained models, we analyze the variations in their outputs for a given crystal structure under perturbations.

The results in Figure A4 show that the weighted PCA frame model exhibits a significantly smoother transition compared to the max frame model. However, as shown in Tables 1 and 2, the weighted PCA frame method has lower performance, indicating that higher continuity does not necessarily translate to better performance. The discontinuous behavior of max frames may have facilitated the early stabilization of frames during training, as discussed in Appendix F.3, leading to the superior performance.

Meanwhile, the discontinuity of the max frame model becomes more significant with larger perturbations. This trend suggests that the model may have limited generalization to out-of-domain data. The technique of weighted frames proposed by Dym et al. (2024) could be applied to improve the continuity of our max frame models.

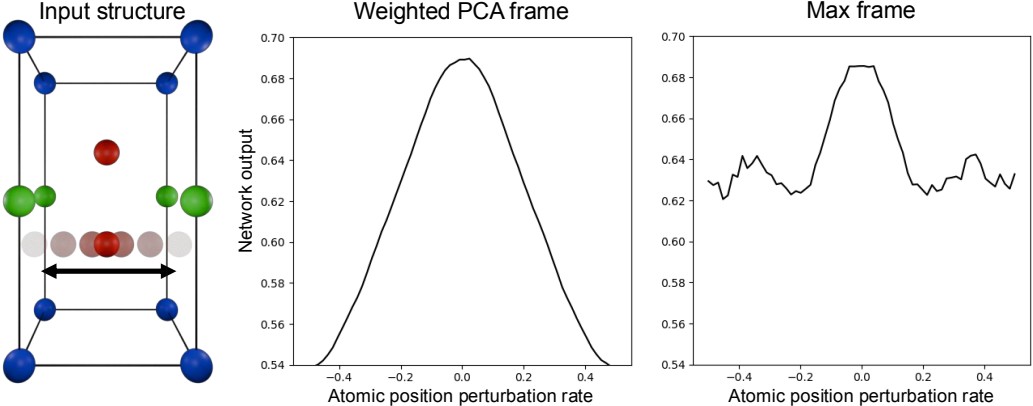

Figure A4: **Continuity under perturbations.** We examine the transitions in the outputs of trained models for a test material under perturbations. Specifically, we use Be$_2$InPb (JVASP-70556) from the JARVIS formation energy prediction task and perturb one of the beryllium (Be) atoms along the direction of a lattice vector. The weighted PCA frame model shows a smoother transition compared to the max frame model.

