# OpenReview forum: "Rethinking the role of frames for SE(3)-invariant crystal structure modeling"
_ICLR.cc/2025/Conference — ICLR 2025 Poster_

### Official Review · Reviewer_dkNi · 2024-10-21

**Soundness:** 3
**Presentation:** 3
**Contribution:** 3
**Rating:** 8
**Confidence:** 3

**Summary:**

This work explores an alternative, more powerful version of frame averaging networks and applies it to crystal property prediction tasks. Instead of using a fixed lattice frame or one calculated from the PCA of the moment tensor of the atoms, the authors propose to learn "dynamic frames" for each individual atom. They propose two methods to do this: Either by calculating the PCA from weighted atoms or constructing the frames by greedily choosing the highest weighted atoms, where the weights in both cases are equal to the attention weights in their transformer architecture. They show that this results in significant performance gains and that the model exhibits intuitive, sensible behaviour, where the earlier layers attend to closer atoms, and later layers attend to further away atoms.

**Strengths:**

Crystal property prediction is an important topic, and (stochastic) frame averaging is a promising direction for the task. Using the novel idea of dynamic frames, the proposed method improves stochastic frame averaging for crystal property prediction by a large margin. Therefore, the contribution seems significant and novel. It is believable that the proposed "dynamic frames" make the model more expressive, although I find it hard to get an intuitive understanding of what the angles between the learned frames and direction vectors could learn or why they could represent useful features. The experimental results show strong improvements on two datasets and against the base model (Crystalformer) and other state-of-the-art models.

**Weaknesses:**

The biggest weakness of this paper is, in my opinion, the writing. Many sentences are grammatically incorrect or use strangely constructed, overly long sentences. This makes it much harder to read the paper than it should be. A non-exhaustive list of examples:

Line 121: "... are linear projects of current state x, σi is a tail-length variable
of Gaussian distance-decay attention adaptively derived from xi, ψij(n) is geometric position
embedding that encodes interatomic". This misses several articles, and I assume the authors mean "projections" not "projects"

Line 198: "... and local view of entire crystal ..." misses the article "the"

Line 234: "This perturbation scheme is considered to implement the stochastic FA" I am not sure what this sentence means

I strongly recommend that the authors use a grammar tool like Grammarly or an LLM to help streamline their writing.

**Questions:**

In the runtime comparison, Table 4: How can PotNet take only 43s/epoch to train but 313ms for inference, while CrystalFramer takes almost double the time to train with 74s but is nearly ten times faster during inference? Can you explain this large discrepancy?

In line 298: The model uses the cosine of the angles. This would map pi/2 and -pi/2 to the same value 0. Have you considered using a concatenation of both sine and cosine of the angles as features? It might make the model more expressive.

---

> ### Author Response · Authors · 2024-11-14
> **Point-by-point responses to Reviewer dkNi (1/2)**
>
> Dear Reviewer dkNi,
>
> Thank you very much for providing us with valuable feedback. We appreciate the detailed comments, and we understand that Reviewer dkNi’s primary concerns are related to the clarity and writing of our paper. Below, we have provided point-by-point responses to each of your comments.
>
> ---
> ## Regarding paper writing
> ### Overall presentation
> > - The biggest weakness of this paper is, in my opinion, the writing. Many sentences are grammatically incorrect or use strangely constructed, overly long sentences. This makes it much harder to read the paper than it should be.
> > - I strongly recommend that the authors use a grammar tool like Grammarly or an LLM to help streamline their writing.
>
> Thank you for your comments. We respectfully note that **the other three reviewers rated the presentation as  "excellent" (4) and "good" (3) with positive comments, such as “presented with commendable clarity” (Reviewer AkN2) and “The paper is well written” (Reviewer SjyW). Based on this feedback, we believe our writing is generally clear and unlikely to be a reason for rejection.**
>
> However, we fully agree that clarity is essential for reaching a broad audience. **We have revised texts throughout the paper to enhance overall readability (see `green parts`).** We will continue to improve our manuscript by carefully reviewing each sentence, fixing typos, simplifying complex constructions where possible, and using AI tools to enhance readability and precision.
>
> ### Omission of articles
> > - Line 121: "... are linear projects of current state x, σi is a tail-length variable of Gaussian distance-decay attention adaptively derived from xi, ψij(n) is geometric position embedding that encodes interatomic". This misses several articles
> > - Line 198: "... and local view of entire crystal ..." misses the article "the"
>
> Thank you for your comments regarding article usage (i.e., "a," "an," and "the"). We intentionally omit articles for mathematical objects (e.g., $x$) and nouns directly preceding mathematical objects (e.g., entire crystal structure $\hat{P}$). In previous feedback from a professional paper editing service, we were advised to follow these specific article usage styles:
> - Function $f$ is continuous.
> - The function, $f$, is continuous.
> - $f$ is continuous.
>
> Following these styles, we believe that when "function $f$" acts like a proper noun, omitting an article (a or the) is acceptable. This style is also supported in public discussions, such as this reference: https://english.stackexchange.com/questions/244182/definite-article-in-maths-the-function-f
>
> Thus, we consider the omission of articles here to be a stylistic choice, reflecting the authors' writing style. However, if Reviewer dkNi (or other reviewers) still find this approach problematic, we are willing to revise the manuscript accordingly.
>
> ----
> CONTINUE

---

> > ### Comment · Reviewer_dkNi · 2024-11-22
> >
> > After rereading the paper, I apologize for the harsh score. I am unsure why I thought it was presented so poorly and you are right about the articles being a stylistic choice. I also think the added explanations and rewriting added more clarity. I am revising my score.

---

> ### Author Response · Authors · 2024-11-14
> **Point-by-point responses to Reviewer dkNi (2/2)**
>
> ---
> ## Technical clarification
>
> ### Intuitive understanding
> > … I find it hard to get an intuitive understanding of what the angles between the learned frames and direction vectors could learn or why they could represent useful features.
>
> Thank you for raising this concern. A dynamic frame is a local coordinate system centered on each atom and aligned with its surrounding dynamic environment. Intuitively, its first and second axes (i.e., e1 and e2) point toward the primary and secondary directions of interatomic influence around the central atom, while the third axis is automatically determined as e3 = e1 x e2. **The angles between these axes and j(n)'s direction vector quantify the angular deviation of the interacting atom j(n) from these influential directions.**
>
> In Section 6 ("Visual analysis"), we visually analyze how angle information can serve as useful features. Specifically, we consider that **these frame angles capture distortions in distinctive local structural patterns**, such as octahedra and tetrahedra, which frequently appear in crystal structures.
>
> **We have revised our paper to convey an intuitive interpretation to Eq 7** in the `blue part blow Eq 7`.
>
> ---
> ### Relationship to stochastic FA
> > Line 234: "This perturbation scheme is considered to implement the stochastic FA" I am not sure what this sentence means
>
> Thank you for pointing this out. Stochastic frame averaging (stochastic FA) is a technique detailed in lines 153-156 of Section 2.3. While the original FA takes the average of the function values across all possible frames (see Eq 4), the stochastic FA efficiently approximates this process by randomly selecting one frame from the possible set.
>
> In our case, when multiple atoms have the same weight, the max frame can select either atom to define a frame axis. This results in multiple possible frame constructions. **Adding small noise to weights forces the choice of a specific frame among these possibilities, making this approach equivalent to stochastic FA.**
>
> **The revised paper now clarifies that our weight perturbation scheme is a type of stochastic FA that is outlined in Section 2.3**. See the blue part in `"Max frames" in Sec 3.1`.
>
> ---
> ## Questions
> > In the runtime comparison, Table 4: How can PotNet take only 43s/epoch to train but 313ms for inference, while CrystalFramer takes almost double the time to train with 74s but is nearly ten times faster during inference? Can you explain this large discrepancy?
>
> Thank you for your question. As noted in Table 4, the per-material test time includes data preprocessing, such as graph construction, whereas this preprocessing time is excluded from the per-epoch training time. In the case of PotNet, its graph construction stage includes edge feature calculations based on periodic interatomic potential summations, which slow down the inference speed. In contrast, CrystalFramer requires less computationally intensive preprocessing, allowing for faster inference.
>
> **The revised paper clarifies that the slower inference times of existing methods are due to their relatively heavy data preprocessing** at `the end of Sec 5.2` (blue part).
>
> ---
> > In line 298: The model uses the cosine of the angles. This would map pi/2 and -pi/2 to the same value 0. Have you considered using a concatenation of both sine and cosine of the angles as features? It might make the model more expressive.
>
> Thank you for this suggestion. The cosine-based angle representation arises naturally from a coordinate transformation, where a 3D vector is projected onto the frame coordinate system as $\vec{\theta} = F_i \vec{r}$. (Note that $F_i$ has an *orthonormal basis*, so $F_i = [e_1, e_2, e_3]^{T} = [e_1, e_2, e_3]^{-1}$ maps from the given coordinate system to the frame coordinate sytem spanned by the three basis vectors: $e_1, e_2, e_3$.)
>
> This cosine-based representation follows prior work such as iComFormer. In our case, it is intended to capture the absolute deviation from the frame axes, which represent influential directions of interactions. However, we agree that concatenating sine and cosine could potentially enhance the model's expressive power, and we consider this an interesting idea to explore in future work.
>
> **The revised paper clarifies that $F_i \vec{r}$ is the projection onto the frame coordinate system and it follows iComFormer** in `Sec 3.2 (blue part above Eq 7)`.
>
> ---
> ---
> Finally, we would like to thank Reviewer dkNi once again for these valuable comments. We will reflect these comments in the revised manuscript. We believe that our responses above address all of Reviewer dkNi's concerns and contribute to further strengthening our work.
>
> Sincerely,
> The Authors

---

> ### Author Response · Authors · 2024-11-19
> **Summary of changes related to Reviewer dkNi's comments**
>
> Dear Reviewer dkNi,
>
> We have carefully revised the manuscript and have just updated the PDF. The changes related to the comments by Reviewer dkNi are summarized below.
>
> *Note: We have released the second revision and updated the reference numbers in our responses accordingly.*
>
> - **Revised texts throughout to ehnance readability** by simplifying complex sentence constructions, fixing typos, etc. These changes are highlighted in green.
> - **Added an intuitive interpretation to frame-based edge features** in `Sec 3.2 (right after Eq 7)` (highlighted in blue).
> - **Clarified that the perturbation scheme is a type of stochastic FA that is outlined in Sec 2.3** in `"Max frames" in Sec 3.1` (highlighted in blue).
> - **Clarified the reason for slower inference speeds of existing methods** (i.e., “relatively heavy data preprocessing”), at the `end of Sec 5.2` (highlighted in blue).
> - **Clarified that the cosine-based angle representation represents the projection onto the frame coordinate system and follows existing work (Yang et al., 2024)** in `Sec 3.2 (above Eq 7)` (highlighted in blue).
>
> Given these improvements, we feel the clarity of our paper has been much improved. We thank Reviewer dkNi for the detailed feedback.
>
> Best Regards,
> The Authors

---

> > ### Comment · Reviewer_dkNi · 2024-11-22
> >
> > Thank you for the clarifications and explanations; I think the paper is in a good state now

---

> > > ### Author Response · Authors · 2024-11-22
> > > **Thank you for your response**
> > >
> > > Dear Reviewer dkNi,
> > >
> > > Thank you very much for your comments and for raising the score. Your valuable feedback has been instrumental in improving our paper.
> > >
> > > Best,
> > > The Authors

---

### Official Review · Reviewer_SjyW · 2024-11-03

**Soundness:** 3
**Presentation:** 3
**Contribution:** 4
**Rating:** 8
**Confidence:** 4

**Summary:**

The paper proposes a new method for modeling crystal based on dynamic frames that are based on the local neighborhood of each atom in the crystal structure. The crystal frames are then processed by a transformer based architecture called CrystalFramer. The paper starts by motivating the utility of geometric deep learning methods that can capture relevant physical inductive biases, such as invariance. Previous work captured such geometric information using frame averaging, which is based on taking canonical frames of the entire structure. By contrast, the paper proposes dynamic frames which are based on the local coordinate system of a given atom. Section 2 then outlines relevant preliminary details with relevant mathematical definitions, including the description of a crystal structure, how transformers have been applied to crystal structure modeling and prior work on frame averaging for molecular and materials modeling.

Section 3 introduces and defines the concept of dynamic frames, which is the main part of the proposed new method. This includes the frame definitions in Section 3.1 and the CrystalFramer architecture in Section 3.2 that is based on Gaussian basis functions. This is followed by a description of related work on invariant features, frames and equivariant features in Section 4.

Section 5 outlines the main experiments in the paper, including materials property prediction and an efficiency analysis . The results generally show that the proposed dynamic frames improve Crystalformer modeling performance across all datasets while achieving best performance in most cases. The efficiency analysis shows that CrystalFramer incurs an extra compute cost. Section 6 provides a visual analysis of the dynamic frames, as well as discussion on limitations along with suggestions for using dynamic frames for equivariant prediction and molecular modeling.

**Strengths:**

* The idea of dynamic frames proposed in the paper is a novel and effective way to include relevant inductive biases for crystal structure modeling.
* The paper is well written with relevant details described and convincing experiments followed by detailed analysis.
* The paper candidly discusses advantages and limitations of the proposed method.

**Weaknesses:**

* The paper could be further with additional discussion of related work for machine learning potentials [1 egraff]. The current focus of the study is mainly on properly modeling, so this would probably fit as potential future work. There be advantages and disadvantages of using CrystalFramer in such a context. Similarly, the paper can also discuss potential application to modeling of crystals with surfaces, such as catalyst [2 - OCP].
* The dynamic frames provide a new dimension for analysis for how local neighborhoods interact in geometric deep learning. The paper could be strengthened by discussing the frames across different materials and/or different stages of the training process.

**Questions:**

* Could you talk more about scalability of your method? This would be especially relevant when modeling larger supercell that have more atoms.
* Did you notice an evolution of the dynamic frames as training progressed? Are there consistencies across some types of materials?

---

> ### Author Response · Authors · 2024-11-19
> **Point-by-point responses to Reviewer SjyW (1/2)**
>
> Dear Reviwer SjyW,
>
> We highly appreciate your insightful and constructive feedback on our work. We are pleased to receive the highest praise for our contribution from such an expert. Below are point-by-point responses to your suggestions with some additional results. We have also updated the manuscript accordingly.
>
> ---
> ### Application for equivariant task benchmarks
> > The paper could be further with additional discussion of related work for machine learning potentials [1 egraff]. The current focus of the study is mainly on properly modeling, so this would probably fit as potential future work. There be advantages and disadvantages of using CrystalFramer in such a context. Similarly, the paper can also discuss potential application to modeling of crystals with surfaces, such as catalyst [2 - OCP].
>
> Thank you for these constructive suggestions. We assume the references “[1 egraff]” and “[2 - OCP]” refer to the following:
> 1. EGraFFBench: evaluation of equivariant graph neural network force fields for atomistic simulations (https://pubs.rsc.org/en/content/articlehtml/2024/dd/d4dd00027g)
> 2. Open Catalyst Project (https://opencatalystproject.org)
>
> Both of these benchmarks primarily involve equivariant prediction tasks, such as force  [1,2] and relaxed structure [2] prediction. As we discuss in our response to Reviewer bzKy, addressing equivariant tasks would entail exploring several potential approaches and conducting additional large-scale experiments. Thus, we agree with Reviewer SjyW that such an extension would be best suited for future work.
>
> Meanwhile, **we have revised the `“Equivariant prediction” in Sec 6` to incorporate the references to these benchmarks, discussing important applications of equivariant networks** (highlighted in red).
>
> ---
> ### Discussion on frames across different materials
> > - The dynamic frames provide a new dimension for analysis for how local neighborhoods interact in geometric deep learning. The paper could be strengthened by discussing the frames across different materials […]
> > - [Question] Are there consistencies across some types of materials?
>
> Thank you for your suggestions to strengthen our paper. To clarify, **Appendix F already provides visualizations of frames for a different material (as Figure A2) than the one shown in Figure 3, and also discusses a general tendency across materials**. We observe that our model tends to attend to close neighbors in shallow layers and to relatively distant neighbors in deeper layers. **The revised paper clarifies in `Sec 6 (blue parts)` that Appendix F includes these contents.**
>
> ---
> ### Evolution of dynamic frames during training
> > - The paper could be strengthened by discussing the frames across [...] different stages of the training process.
> > - [Question] Did you notice an evolution of the dynamic frames as training progressed?
>
> To respond to this interesting question, **we have added new frame visualizations as `Figure A3` in `Appendix F.3`, showing the evolution of dynamic frames during training**. This new analysis reveals that the weighted PCA frames tend to fluctuate throughout training, whereas the max frames stabilize quickly during the early stages. As frame fluctuations can introduce noise, this difference may explain the superior performance of the max frame method.
>
> ---
> CONTINUE

---

> ### Author Response · Authors · 2024-11-19
> **Point-by-point responses to Reviewer SjyW (2/2)**
>
> ---
> ### Scalability for large structures and supercells
> > [Question] Could you talk more about scalability of your method? This would be especially relevant when modeling larger supercell that have more atoms.
>
> Regarding scalability w.r.t. structure size, CrystalFramer is based on a self-attention mechanism, whose complexity is generally $O(Nk)$, where $N$ is the number of atoms in the unit cell and $k$ is the number of neighbors per unit-cell atom.  In the infinitely connected attention of Crystalformer defined in Eq. 3, neighbors $j(n)$ are adaptively determined for each atom $i$ in each layer. The current implementation computes neighbors by periodically repeating the unit cell within a finite range. Consequently,
> $k$ becomes a multiple of $N$, resulting in **an overall computational complexity of $O(N^2)$**.
>
> In practice, the training of CrystalFramer has successfully scaled to relatively large structures in the MP dataset, which features an average of 30 atoms per unit cell and a maximum of 296 atoms. For inference, the method can handle even larger structures than during training, as it requires significantly less memory and supports per-material (non-batched) processing.
>
> We understand that supercells become crucial when structures deviate from perfect periodicity, such as in the presence of impurities, defects, or surfaces. For such cases, we consider two potential approaches to improve efficiency with large supercells:
>
> 1. **Mixed atom embedding**. Structures with impurities or defects are often represented using site occupancy, which indicates the probabilities of different elements occupying an atomic site. Instead of modeling such structures with supercells, we can efficiently represent the site occupancy by mixing atomic embedding vectors. In this case, each atomic species $a_i$ represents a probability distribution over elements rather than a single element. The corresponding atomic state can then be initialized as a linear blend of atom embeddings: $x_i \gets \sum_{\text{element}} a_i(\text{element}) \text{AtomEmbedding}(\text{element})$. This approach can keep the structure size small without using a supercell, thereby maintaining overall efficiency.
>
> 2. **Distance-based neighbor search**. When unit cells are large, the current cell-based neighbor identification method will produce redundant neighbors, forcing $k \ge N$. By employing a more compact set of neighbors through nearest neighbor search, the complexity is reduced from $O(N^2)$ to $O(Nk)$, improving efficiency for larger structures.
>
>
> We think that developing scalable models for structures with imperfect periodicity is an interesting and practically important direction for future research. **We have revised the manuscript to include a new discussion on scalability in `Appendix G`, and have mentioned it from `Sec 5.2 (red part)`.**
>
> ---
> We thank Reviewer SjyW again for providing these comments. By reflecting the feedback into the revision, we feel that our work has been greatly strengthened.
>
> Sincerely,
> The Authors

---

> ### Author Response · Authors · 2024-11-19
> **Summary of changes related to Reviewer SjyW's comments**
>
> Dear Reviewer SjyW,
>
> We have just updated the PDF. The changes related to the comments by Reviewer SjyW are summarized below.
>
> *Note: We have released the second revision and updated the reference numbers in our responses accordingly.*
>
> - **Added a duscussion on important applications of equivariant extensions and new references to EGraff and Open Catalyst Project benchmarks** in  `"Equivariant prediction" in Sec 6` (highlighted in red).
> - **Added a new analysis of frame evolution during training in new `Appendix F.3`** (highlighted in red) and mentioned it from `"Visual analysis" in Sec 6`  (highlighted in blue).
> - **Added a new discussion on scalability for large structures in new `Appendix G`** and mentioned it from `Sec 5.2` (highlighted in red).
> - **Clarified that Appendix F includes additional frame visualizations and comparative discussions** in `"Visual analysis" in Sec 6` (highlighted in blue).
> - Additionally, the texts in the paper have been polished in response to Reviewer dkNi's comment to improve readability. (These changes are highlighted in green)
>
> We believe that these new results and dicussions have greatly strengthened the overall quality of our paper.
> Thank you again for suggesting them.
>
> Best Regards,
> The Authors

---

> > ### Comment · Reviewer_SjyW · 2024-11-22
> >
> > I appreciate the updates and the discussion of equivariance as in the context of the method, as well as the additional inclusion of frame evolution. I think the new revisions make the paper stronger.

---

> > > ### Author Response · Authors · 2024-11-22
> > > **Thank you for your response**
> > >
> > > Dear Reviewer SjyW,
> > >
> > > Thank you very much for taking the time to confirm our responses. We sincerely appreciate your recognition of our work and your valuable suggestions, which have greatly contributed to strengthening our paper.
> > >
> > > Best,
> > > The Authors

---

### Official Review · Reviewer_bzKy · 2024-11-04

**Soundness:** 3
**Presentation:** 3
**Contribution:** 3
**Rating:** 6
**Confidence:** 3

**Summary:**

This paper proposes CrystalFramer and dynamic frames to capture SE(3)-invariant geometric features for crystal property prediction. Unlike using static frame in previous work, this work use dynamics frames to give each atom its own dynamics view of the structure, focusing only on those atoms actively interacting with it, meanwhile also accommodating the infinite and symmetric nature of crystal structures. The experiment shows superior performance of the proposed method in several prediction tasks.

**Strengths:**

- This work proposes dynamic frames for crystal features modeling, which can dynamically accounting for the atoms actively engaged in learned interactions in each interatomic message-passing layer.
- The proposed method achieves superior performance on several commonly used benchmarks in crystal property prediction.

**Weaknesses:**

- Previous relevant frame-based method ComFormer also provides equivariant version. I am wondering if this method can also be easily extended to SE(3)-equivariant? There’re are many other equivariant properties for the materials such as force or high-order tensors. I would like to see the performance of equivariant version of this method if the extension is straightforward. Even for the invariant properties, it’s interesting to know whether equivariant network can help invariant properties prediction.

**Questions:**

- Please refer to the weakness part.

---

> ### Author Response · Authors · 2024-11-15
> **Responses to Reviewer bzKy**
>
> Dear Reviewer bzKy,
>
> Thank you very much for recognizing the value of our work and providing constructive feedback. Below, we respond to your comments on the application for equivariant property prediction.
>
> ---
> ## Equivariant prediction
> > Previous relevant frame-based method ComFormer also provides equivariant version. I am wondering if this method can also be easily extended to SE(3)-equivariant? There’re are many other equivariant properties for the materials such as force or high-order tensors. I would like to see the performance of equivariant version of this method if the extension is straightforward. Even for the invariant properties, it’s interesting to know whether equivariant network can help invariant properties prediction.
>
> We agree that extending our work for equivariant property prediction would be both interesting and valuable. In the “Equivariant prediction” of Sec 6, we discuss two potential approaches for this extension, one similar to the original FA, and the other similar to Graphormer3D (Shi et al., 2023).
>
> However, thoroughly evaluating these approaches on several equivariant tasks would require extensive experimentation, combining {tasks} x {existing frames + proposed frames} x {two potential approaches}, resulting in a large number of additional experiments. Given our current time and computational constraints, we believe these investigations would be best suited as a separate study or as an extended work for a journal submission.
>
> > Previous relevant frame-based method ComFormer also provides equivariant version.
>
> Let us clarify that **the work of ComFormer (Yan et al., 2024) also focuses only on invariant property prediction tasks**, as our work does. Although eComFormer leverages equivariant features internally within “equivariant updating layers”, the outputs of these layers, as well as the final output, are invariant. In fact, its extension for equivariant tensor prediction has recently appeared in a separate study (https://openreview.net/forum?id=0k7pbSxNOG). These examples suggest that **tackling both invariant and equivariant tasks in a single study is challenging**, especially given the recent demand for extensive evaluations and the complexities of equivariant tasks and crystal-structure modeling.
>
> ---
> Overall, we believe that **exploring applications to equivariant tasks would be more appropriate for future research, and focusing on invariant tasks would not hurt the value of our research**. We understand that this perspective is also supported by
> Reviewer SjyW, who noted that the EGraFFBench and Open Catalyst Project (ie, equivariant task benchmarks) “would probably fit as potential future work”.
>
> **The revised paper incorporates references to these equivariant-task benchamrks and dicusses them  in `"Equivariant prediction" in Sec 6`** as important pontential applications.
>
> We recognize that Reviewer bzKy’s comments highlight these important future directions, and we thank once again for the insightful and constructive feedback.
>
> Sincerely,
> The Authors

---

> ### Author Response · Authors · 2024-11-19
> **Summary of changes related to Reviewer bzKy's comments**
>
> Dear Reviewer bzKy,
>
> We have revised the manuscript and have just updated the PDF.
>
> - **Added a discussion on important applications of equivariant extensions and new references to benchmarks on equivariant tasks** in `"Equivariant prediction" in Sec 6` (highlighted in red), in response to both Reviewer bzKy and Reviewer SjyW.
> - Additionally, the texts in this paragraph, along with other sections, have been polished in response to Reviewer dkNi's comment to improve readability. (These changes are highlighted in green)
>
> Best Regards,
> The Authors

---

### Official Review · Reviewer_AkN2 · 2024-11-04

**Soundness:** 3
**Presentation:** 4
**Contribution:** 3
**Rating:** 6
**Confidence:** 4

**Summary:**

The authors change the fixed, static frames for SE(3)-invariant modeling in crystal structures into dynamic frames that adapt to each atom's local environment through the network's attention mechanism, then integrate into a new architecture **CrystalFramer**. Experimental results on large-scale datasets (JARVIS, MP, OQMD) demonstrate that **CrystalFramer** with dynamic frames outperforms models using static frames in several material property prediction tasks.

**Strengths:**

1. The proposed method is straightforward and presented with commendable clarity.
2. The shift from static to dynamic frames enables atom-specific adjustments, significantly enhancing SE(3) invariance handling in highly symmetric crystal structures. Furthermore, the extensive evaluations on multiple datasets (JARVIS, MP, OQMD) highlight the effectiveness of dynamic frames, with the "max frames" variation demonstrating particular superiority.

**Weaknesses:**

### Major
How do the "max frame" construction and its usage within the networks differ from those proposed in [1], [2], [3], and [4]? In particular, [1] and [4] use similar atomic local frames to address equivariance in molecular structures. A comparative discussion of these methods would be valuable to clarify whether the proposed approach is a direct adaptation from molecular modeling to materials or if there are novel insights beyond those methods.

### Minor
Using frame-based techniques will hinder the model's continuity, potentially affecting its generalization capabilities, as noted in [5]. Does the proposed model address this limitation? It will be valuable to discuss the continuity of the proposed dynamic frame approach and empirically analyze if this affects generalization in the experiments.



[1] "SE (3) equivariant graph neural networks with complete local frames." Du, et al.

[2] "Smooth, exact rotational symmetrization for deep learning on point clouds." Pozdnyakov, Sergey, and Michele Ceriotti.

[3] "Equivariance via Minimal Frame Averaging for More Symmetries and Efficiency." Lin, et al.

[4] "A new perspective on building efficient and expressive 3D equivariant graph neural networks." Du, et al.

[5] "Equivariant frames and the impossibility of continuous canonicalization." Dym, Nadav, Hannah Lawrence, and Jonathan W. Siegel.

**Questions:**

See weaknesses.

---

> ### Author Response · Authors · 2024-11-20
> **Point-by-point responses to Reviewer AkN2 (1/2)**
>
> Dear Reviewer AkN2,
>
> We sincerely appreciate Reviewer AkN2's valuable feedback. We are also honored to receive such high praise regarding the clarity. In response to the review, **we have prepared point-by-point responses, introduced new experimental results, and revised the manuscript accordingly**.
>
> ---
> ## Dynamic frames vs existing local frames for molecules
> > How do the "max frame" construction and its usage within the networks differ from those proposed in [1], [2], [3], and [4]? In particular, [1] and [4] use similar atomic local frames to address equivariance in molecular structures. A comparative discussion of these methods would be valuable to clarify whether the proposed approach is a direct adaptation from molecular modeling to materials or if there are novel insights beyond those methods.
>
> Thank you for pointing out these references. To begin with the conclusion, **the concept of our dynamic frames, being both dynamic and local, is clearly distinct from these existing frames for molecules, which are local but static**.
>
> To prevent potential confusion, we clarify that “dynamic” in this context refers to behavior that is influenced by models' internal states estimated for a given structure. For instance, interatomic interactions modeled within a GNN reflect these internal states and evolve dynamically layer by layer. Dynamic frames are designed to align with these interatomic interactions. While the molecular modeling literature often uses “dynamic” to describe temporally evolving structures, our work does not assume such temporal dynamics. Similarly, we use “static” to describe behavior that is unaffected by models' internal states.
>
> Below, we provide comments on each of [1, 2, 3, 4] and present new experimental results to support our perspective.
>
> ---
> ### Literature review (see Appendix B for a detailed version)
> 1. **Du et al. (2022) [1] propose static edge-wise frames** constructed with $p_i$, $p_j$, and the structure centroid $\overline{p}$. However, determining centroids for crystal structures is not straightforward due to their infinite nature.
> 2. **Pozdnyakov & Ceriotti (2023) [2] propose ensemble of many 3-body interactions using static triplet-wise frames**. However, modeling 3-body interactions is computationally expensive.
> 3. **Lin et al. (2024) [3] propose minimal frame averaging (minimal FA) to improve the efficiency of FA.** While the stochastic FA used in our method shares a similar goal, the minimal FA ensures exact equivariance (or invariance). Our method could incorporate the minimal FA.
> 4. **Du et al. (2023) [4] propose frame-based equivariant message passing using static edge-wise [1] and node-wise frames**, where node-wise frames are constructed with $p_i$, the structure centroid $\overline{p}$, and the cluster centroid $\overline{p}_i$ around $i$. However, the high symmetry of crystal structures will often cause $\overline{p_i} \simeq p_i$, resulting in unstable frame construction.
>
> Overall, these existing methods either use specific types of static local frames (i.e., node-wise [2], edge-wise[1, 4], or triplet-wise [2]) or propose a mathematical framework for efficient FA [3]. **None of them introduce the concept of dynamic frames leveraging models' internal states.**
>
> ---
> ### Experimental comparison
> To further highlight the conceptual difference between our dynamic frames and the existing frames [1, 2, 4], we developed a variant of CrystalFramer using static local frames. These frames are similar to max frames but constructed with static weights, $w_{ij(n)} = \exp(-r_{ij(n)}^2)$. Thus, they rely solely on the distances to neighbors and do not account for dynamic self-attention weights.
>
> Below, we present benchmark results, comparing the dynamic frame method (max frames) with its static counterpart (static local frames). The results clearly demonstrate the superiority of the proposed dynamic frames over static local frames.
>
> |(JARVIS dataset)|Form. E.|Total E.|BG (OPT)|BG (MBJ)|E hull|
> |-|-|-|-|-|-|
> | Static local frames |0.0285 |0.0292|0.122|0.261| **0.0444** |
> | Max frames| **0.0263** | **0.0279** | **0.117** | **0.242** | 0.0471 |
>
> |(MP dataset)|Form E.|BG|Bulk mod.|Shear mod.|
> |-|-|-|-|-|
> | Static local frames |0.0178 | 0.191 | 0.0354 | 0.0708 |
> | Max frams| **0.0172** | **0.185** | **0.0338** | **0.0677** |
>
> ---
> ### Summary
> - The suggested frame-based methods [1, 2, 4] use specific types of static local frames, which are conceptually distinct from our dynamic frames.
> - The new experimental comparisons demonstrate the superiority of dynamic frames over static local frames.
> - The minimal FA [3] could be integrated into our method as an alternative to stochastic FA to ensure exactness.
>
> **The revised manuscript discusses these related works [1, 2, 3, 4] in `Sec 4` and new `Appendix B`, and presents the new comparisons in `Sec 5.1` with the updated `Tables 1 and 2`.** (All highlighted in red)
>
> ---
> CONTINUE

---

> ### Author Response · Authors · 2024-11-20
> **Point-by-point responses to Reviewer AkN2 (2/2)**
>
> ---
> ### Model’s continuity
> > Using frame-based techniques will hinder the model's continuity, potentially affecting its generalization capabilities, as noted in [5]. Does the proposed model address this limitation? It will be valuable to discuss the continuity of the proposed dynamic frame approach and empirically analyze if this affects generalization in the experiments.
>
> Our approach inherits the discontinuity of frame-based models. **To empirically assess the degree of this discontinuity in our trained models, we analyzed the variations in their outputs for a given crystal structure under perturbations.**
>
> The results in Figure A4 (Appendix H) show that the weighted-PCA frame model exhibits a significantly smoother transition compared to the max frame model. However, as shown in Tables 1 and 2, the weighted-PCA frame method has lower performance, indicating that **higher continuity does not necessarily translate to better performance**. The discontinuous behavior of max frames may have facilitated the early stabilization of frames during training, as discussed in Appendix F.3, possibly contributing to the superior performance.
>
> Meanwhile, the discontinuity of the max frame model becomes more significant with larger perturbations. This trend suggests that the model may have limited generalization to out-of-domain data. The technique of weighted frames [5] could be applied to improve the continuity of our max frame models.
>
> **The revised paper includes this new continuity analysis in `Appendix H`, and mentions it along with the frame evolution analysis in the `“Visual analysis” of Sec 6`.** (Highlighted in red)
>
> ---
> We thank Reviewer AkN2 once again for providing these insightful references, which have greatly helped us deepen our understanding and better contextualize our work. We believe that the new discussions, comparisons, and analyses have strengthened our work and will address all of Reviewer AkN2's concerns.
>
> Sincerely,
> The Authors

---

> ### Author Response · Authors · 2024-11-20
> **Summary of changes related to ReviewerAkN2's comments**
>
> Dear Reviewer AkN2,
>
> We have just updated the PDF. The changes related to the comments by Reviewer AkN2 are summarized below. These changes are **highlighted in red** in the paper.
>
> - **Added a new comparative discussion on frame-based methods for molecules in new `Appendix B`** and in `the last paragraph of Sec 4`.
> - **Added new benchmark results for a static local frame method** in `Tables 1 and 2` and `Sec 5.1` as a static counterpart to the (dynamic) max frame method.
> - **Added a new reference to minimal FA [5]** in `"Frames" in Sec 4`.
> - **Added a new analysis on model's continuity in new `Appendix H`** and in `"Visual analysis" in Sec 6`.
> - Additionally, the texts in the paper have been polished in response to Reviewer dkNi's comment to improve readability. (These changes are highlighted in green)
>
> Best Regards,
> The Authors

---

> > ### Comment · Reviewer_AkN2 · 2024-11-22
> >
> > Thank you for your rebuttal. I believe frame construction via invariant neural weights is a good idea and the paper is in good shape. I increase the score of contributions and the rating accordingly.

---

> > > ### Author Response · Authors · 2024-11-22
> > > **Thank you for your response**
> > >
> > > Dear Reviewer AkN2,
> > >
> > > Thank you very much for your comments and for raising the score. Your feedback was essential in deepening and better contextualizing our work.
> > >
> > > Best,
> > > The Authors

---

### Author Response · Authors · 2024-11-13
**The authors are preparing point-by-point responses to each reviewer**

Dear Reviewers and Meta-Reviewers,

We sincerely appreciate the time and effort you have put into reviewing our work. We are pleased to receive the positive feedback, including "excellent" ratings for both presentation and contribution. We are confident that the remaining concerns raised by Reviewers AkN2 and dkNi will be addressed in our forthcoming responses, which we are currently preparing.

We will provide point-by-point responses to each Reviewer within a few days.
Thank you again for your valuable feedback and commitment.

Sincerely,
The Authors

---

### Author Response · Authors · 2024-11-22
**Request for Reactions from Reviewers**

Dear All Reviewers and Area Chairs,

We have responded to all the Reviewers' comments in their respective threads and updated the manuscript accordingly. As the end of the rebuttal period is approaching, we sincerely ask all Reviewers for their reactions, such as adjusting scores or posting follow-up questions. Below, we summarize the **main concerns and our responses** to assist Reviewers and Area Chairs.

---
### **Main Concerns and Our Responses**
- **Novelty Compared to Existing Local Frames for Molecules** (Reviewer AkN2):
  - Our work introduces **a clear conceptual difference** by proposing *dynamic frames* that account for models' internal states, whereas existing *static frames* do not. (See `Appendix B` for detailed discussions.)
  - Additional experiments validate the **superiority of (dynamic) max frames over a static counterpart using static local frames** (see `Tables 1 and 2` and  `Sec 5.1`).
- **Clarity and Writing** (Reviewer dkNi):
  - Three reviewers rated the presentation as "excellent" (4) and "good" (3) with positive comments, such as “presented with commendable clarity” (Reviewer AkN2) and “paper is well written” (Reviewer SjyW)
  - We have **revised texts throughout to further enhance readability** (see `texts highlighted in green`).
- **Application to Equivariant Tasks** (Reviewer bzKy & Reviewer SjyW):
  - This was raised as **a constructive suggestion** rather than a critical weakness by two positive reviewers.
  - However, **addressing both invariant and equivariant tasks in a single study is challenging**, given the complexities of equivariant tasks and crystal-structure modeling as well as the recent demand for extensive evaluations.
  - Prior works on crystals (even the mentioned eComFormer) also focus solely on invariant tasks.
  - We have incorporated **references to equivariant-task benchmarks (Open Catalyst and EGraFFBench) for future research**, as suggested by Reviewer SjyW. (See `"Equivariant prediction" in Sec 6`)

---
### **Summary of Updates**
**Highlight colors** in the revised manuscript are intended as follows
- `Red`: Major changes introducing **new information**, such as additional references, discussions, and results.
- `Blue`: Relatively minor changes to enhance **clarity and understanding**, including improved interpretations and corrections of inaccuracies.
- `Green`: Minor changes made to improve **overall readability** (e.g., breaking up long sentences), addressing the comments from Reviewer dkNi.

**◆ Major Changes** (red highlights)
- New comparative discussions on **existing local frames for molecules** (Du et al., 2022; 2023;
Pozdnyakov & Ceriotti, 2023) in **`Appendix B`** and `Sec 4 (last paragraph)`. [Reviewer AkN2]
- New **benchmark results for a static local frame method** in `Tables 1 and 2` and `Sec 5.1`.  [Reviewer AkN2]
- New experimental analysis on **model's continuity** in **`Appendix H`** and `"Visual analysis" in Sec 6`.  [Reviewer AkN2]
- New visual analysis on **frame evolution during training** in **`Appendix F.3`** and `"Visual analysis" in Sec 6`.  [Reviewer SjyW]
- New dicussions on **scalability for large structures** in **`Appendix G`** with reference from `Sec 5.2`.  [Reviewer SjyW]
- New references to **equivariant task benchmarks (Open Catalyst and EGraFFBench)** as important potential applications in `"Equivariant prediction" in Sec 6`.  [Reviewer bzKy & Reviewer SjyW]
- New reference to **the Minimal Frame Averaging framework (Lin et al., 2024)** in `"Frames" in Sec 4`.  [Reviewer AkN2]

**◆ Minor and Other Changes** (blue and green highlights)
- **Enhanced readability** throughout the paper (green parts) [Reviewer dkNi]
- Added **intuitive interpretations of the frame-based edge features (Eq 7)** in `Sec 3.2`.  [Reviewer dkNi]
- Clarified **the reason for slower inference times of existing methods** at `the end of Sec 5.2`. [Reviewer dkNi]
- **Shrinked `Tables 1 and 2`** and moved the full results to `Appendix E` (`Tables A2 and A3`)
- *Clarified the contents of Appendix F* in `"Visual analysis" in Sec 6`. [Reviewer SjyW]
- *Clarified the relation between the perturbation sheme and stochastic FA* in `"Max frames" in Sec 3.2`. [Reviewer dkNi]
- *Corrected the description of lattice frames* in `"Sec 2.3`.
- *Corrected the architectural description (i.e., final "linear" → "feed forward")* in `"Overall architecture" in Sec 3.2` and `Figure 2`.

---
Given these updates, we feel the depth and quality of our paper have been greatly enhanced. We believe that these revisions, together with our responses in the review threads, adequately address all concerns raised by the Reviewers.

We sincerely appreciate your time and thoughtful feedback.

Best Regards,
The Authors

---

### Meta-Review · Area_Chair_VZKk · 2024-12-13

**Metareview:**

The authors presented a new architecture CrystalFramer for frame-based SE(3)-invariant crystal structure modeling, which introduces new dynamic local frames instead of existing static frames to better capture symmetry and equivariance in crystal materials with experiments showing better crystal property prediction performances.

The proposed dynamic frame as well as corresponding attention mechanisms and invariance neural network for message passing is an interesting methodological contribution for crystal materials modeling, with strong empirical results.

**Additional Comments On Reviewer Discussion:**

During rebuttal responses and discussions, the authors have provided clarifications and significant ablation study results to demonstrate the contributions. The revised manuscript based on these discussions have significantly improved the manuscript.

---

### Decision · Program_Chairs · 2025-01-22

Accept (Poster)